# Adapting Monte Carlo Tree Search for Generative Flow Network Training

## Abstract

Generative Flow Networks, or GFlowNets, formulate generative modelling in discrete spaces as a sequential decision-making problem. Sampling plays a key role in GFlowNet training, as most algorithms use the learned policy to sample trajectories from the environment. Monte-Carlo Tree Search (MCTS) is a planning algorithm that has successfully been applied to train sequential decision-making models with reinforcement learning (RL). In this work, we leverage known connections between GFlowNets and maximum-entropy RL to adapt MCTS for GFlowNet training. We prove that standard MCTS tree construction processes can be modified to calculate the optimal flows for a GFlowNet, given sufficient samples from the environment. Our results extend to multiple cases of GFN modelling, including terminating-energy and intermediate-energy environments. We investigate practical strategies for employing MCTS as a sampling tool and apply it to different GFN parameterizations and training objectives. Through extensive experiments in a variety of discrete domains, including a language-based reasoning task, we show that our proposed method offers an improvement over standard on-policy sampling.

## 1 Introduction

Generative Flow Networks, or GFlowNets (Bengio et al., 2021a;b), are generative models that can sequentially generate objects based on their energy (or reward). GFlowNets act as energy samplers such that they learn and match the underlying energy function and can sample from multiple modes. The GFlowNet policy generates a sample sequentially, by taking one action at a time, and from this perspective, is similar to a Reinforcement Learning (RL) policy. However, unlike the RL training objective that mainly focuses on reward maximization, the GFlowNet training objective aims to learn a policy that can match the reward or energy distribution and sample proportionally to it. While Monte-Carlo Markov chains (MCMC) methods can also sample from an unnormalized energy function, they are computationally expensive and slow to achieve mode-mixing. However, GFlowNets amortize the expensive computation in a single trained generative pass, making it possible to leverage the generalization capabilities of machine learning and learn structure in the energy distribution.

The training objective of GFlowNets is usually formulated in the form of a flow objective such that the incoming and outgoing flows are matched (Bengio et al., 2021a;b; Malkin et al., 2022; Madan et al., 2023). The data to compute these objectives is commonly collected using the current GFlowNet policy and the quality of these collected samples can affect the training efficiency of GFlowNets. However, on-policy approaches face several limitations. They can fail to explore the environment by overfitting to high reward (low energy) trajectories that were sampled recently. This is of particular concern in low-entropy environments, where the reward landscape is quite sparse. On-policy approaches also suffer from poor sample efficiency, as each sampled trajectory is only used for a single gradient update. A number of works have taken inspiration from RL methods to improve sampling for GFlowNet training, including $\epsilon$-uniform exploration, replay buffers, and local search (Kim et al., 2024). In this work, we propose a novel and flexible way of improving data sampling and training efficiency of GFlowNets.

Monte-Carlo Tree Search (MCTS) is a widely used planning algorithm in Reinforcement Learning that has been successfully applied to a number of settings (Browne et al., 2012; Silver et al., 2017;

Kajita et al., 2020). Since GFlowNets can be trained in an off-policy manner and usually operate in a deterministic environment dynamics, we demonstrate how integrating MCTS with GFlowNets can improve data sampling, leading to a more efficient learning algorithm. Through extensive experiments over a wide range of tasks, including an LLM based reasoning benchmark, we show how combining MCTS with GFlowNets can enable efficient training throughout the different default training objectives and GFlowNet parameterizations.

The main contributions of our work are the following:

1. We introduce Monte Carlo DAG Search (MCDS), an adaptation of MCTS that can be used to calculate optimal flows in a GFlowNet environment.

2. We provide a method for using the proposed MCDS to influence trajectory sampling in a way that improves GFlowNet training.

3. We empirically demonstrate the effectiveness of our method with different GFlowNet parameterizations and environment structures.

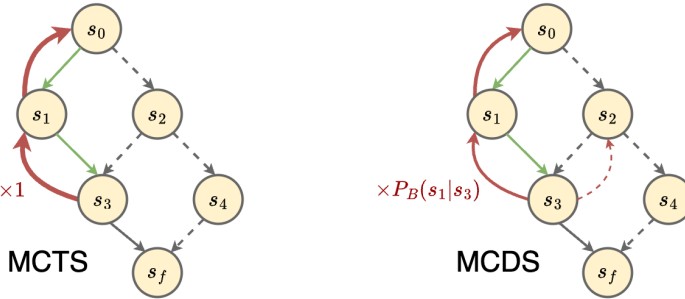

Figure 1: Visual comparison of the Backup operation in Monte Carlo DAG Search (MCDS) vs standard MCTS. In MCDS, the information passed from a child node $s'$ to its parent is modulated by a distribution $P_B(s|s')$ over all parents $s$, taking into account the DAG structure of the environment.

## 2 BACKGROUND

### 2.1 GENERATIVE FLOW NETWORKS (GFLOWNETS)

Consider a Directed Acyclic Graph, or DAG, $G = (\mathcal{S}, \mathcal{A})$ where $\mathcal{S}$ and $\mathcal{A}$ represent the state and action spaces. Given any two states $s \in \mathcal{S}$ and $s' \in \mathcal{S}$, an edge is represented as $(s, s')$. The action space $\mathcal{A}$ consists of directed edges $\mathcal{S} \times \mathcal{S}$ and is thus made up of transitions between any two states. A *trajectory* $\tau$ is represented as a sequence of states: $\tau = (s_1 \rightarrow s_2 \rightarrow \cdots \rightarrow s_n) = (s_1, s_2, \ldots, s_n)$ such that the corresponding actions are $(s_i \rightarrow s_{i+1}) \in \mathcal{A}$ that iteratively build this trajectory one action, or one time step, at a time. Since $G$ is a DAG, there exists no trajectory with $s_n = s_m; \forall n > m$. Given a transition $s_t \rightarrow s_{t+1}$, the state $s_t$ is called the *parent* of $s_{t+1}$ and $s_{t+1}$ is called the *child* of $s_t$. A state $s'$ is said to be a *descendent* of $s$ if there exists a trajectory $\tau \in \mathcal{T}$ such that $s'$ appears after $s$: we denote this relationship as $s \prec s'$. A special initial state, called *source state*, $s_0$, is defined such that $s_0 \prec s$ for all $s \in \mathcal{S} \backslash \{s_0\}$. Similarly, a final state called a *sink state* $s_f$ is defined such that $s \prec s_f$ for all $s \in \mathcal{S} \backslash \{s_f\}$. The parent of a sink state $s_f$ is called a *terminating state*: the set of terminating states is denoted $\mathcal{X} \subseteq \mathcal{S}$. A *complete trajectory* is represented as $\tau = (s_0, s_1, \ldots, s_n, s_f)$ and the set of all complete trajectories is denoted by $T$.

An *environment* is a combination of a state graph $G$ and an energy function $\mathcal{E}(s) : \mathcal{S} \rightarrow \mathbb{R}^+$. The energy can also be expressed as the *reward* function $R(x) = \exp{-\mathcal{E}(x)/\alpha}$, where $\alpha$ is a temperature parameter (typically we set $\alpha = 1$). The reward function can be normalized to define a distribution over terminating states $P(x) \propto R(x)$. The goal of GFlowNet training is learn a model $P(x; \theta)$ that approximates $P(x)$. The environment is described as having *terminating energy* (or terminating reward) if $\mathcal{E}(s) = \infty$ for all $s \in \mathcal{S}/\mathcal{X}$, otherwise it is referred to as *intermediate energy* (intermediate reward).

Given a graph $G = (\mathcal{S}, \mathcal{A})$ as defined above, a *forward policy*, $P_F$, can be defined in terms of a forward probability function $P_F : \sum_{s':s' \in Ch(s)} P_F(s'|s) = 1$, where $Ch(s)$ is the set of children

of $s$. Similarly a *backward policy*, $P_B$, can be defined in terms of a backward probability function $P_B : \sum_{s:s \in Par(s')} P_B(s|s') = 1$.

A state flow $F(s) : \mathcal{S} \to \mathbb{R}^+$ is defined as the measure of the set of complete trajectories passing through a state $s$. An edge flow $F(s, s') : (\mathcal{S} \times \mathcal{S}) \to \mathbb{R}^+$ is defined as the measure of the set of the complete trajectories through an edge $(s, s')$. The *flow* through a set of trajectories $A \subseteq \mathcal{T}$ is defined as the sum of flows of all trajectories $\tau \in \mathcal{A}$. The *total flow* $Z$ is the total flow through all the trajectories $\tau$, defined as: $Z = F(\mathcal{T}) = \sum_{\tau \in \mathcal{T}} F(\tau) = \sum_{x \in \mathcal{X}} R(x)$.

A flow $F$ corresponding to a given graph $G$ is called Markovian if it satisfies the Markovian property, i.e. the next state $s'$ only depends on the current state $s$ and not the previous history. Formally, given a trajectory $\tau = (s_0, s_1, \ldots, s_n, s_f)$, a flow is called Markovian if $\forall (s \to s'), P(s'|\tau) = P(s'|s)$. Using this Markovian flow formulation, a trajectory $\tau$ can be generated by either iteratively sampling the next state $s'$ forward from the current state $s$ using the forward transition probability $P_F(s'|s)$ until reaching $s_f$, or starting at $s_f$ and iteratively sampling the parent state $s$ backwards from the current state $s'$ until $s_0$ is reached.

Many training objectives have been defined for GFlowNets, such as Flow Matching objective (Bengio et al., 2021a), Detailed Balance objective (Bengio et al., 2021b), Trajectory Balance objective (Malkin et al., 2022) and SubTB($\lambda$) objective (Madan et al., 2023), and these operate on the level of the state, edge, full length (complete) trajectories and sub-trajectories of any lengths, respectively. These training objectives are obtained by setting up a set of flow-matching constraints with the property that when all these constraints are satisfied, the GFlowNet sampling policy has the desired property that generates terminating states with probability proportional to $R(x)$. Each constraint can be turned into a loss, typically by taking the square of the logarithm of the ratio of the right-hand side to the left-hand side of the equality constraint. Each loss term thus corresponds to an amount of constraint violation. Training consists in sampling trajectories and measuring these constraint violations (the loss) and its gradient on the parameters of interest.

The Flow Matching (FM) (Bengio et al., 2021a) objective parameterizes GFlowNets through edge flows $F(s \to s'; \theta)$ on states $s$. The Trajectory Balance (TB) (Malkin et al., 2022) objective works with complete trajectories, and parameterizes the GFlowNet through an initial state flow $Z_\theta$, and forward and backward policies, $P_F(s'|s; \theta)$ and $P_B(s|s'; \theta)$, respectively. The Detailed Balance (DB) (Bengio et al., 2021b) and the SubTB($\lambda$) (Madan et al., 2023) objectives parameterize the state flow $F(s; \theta)$, forward policy $P_F(s'|s; \theta)$, and backward policy $P_B(s|s'; \theta)$ on actions $s \to s'$ to define a GFlowNet. The flow-matching constraints represented by these parameterized quantities are converted into a loss function by equating the left and right hand sides of the constraint equations as a squared loss. The flow matching equation for the DB loss is described by Equation 1:

$$F(s; \theta) P_F(s'|s; \theta) = F(s'; \theta) P_B(s|s'; \theta) \tag{1}$$

## 2.2 MONTE-CARLO TREE SEARCH (MCTS)

Although MCTS can in principle be applied to a variety of environments, for simplicity we consider only environments with DAG-structured discrete state spaces, as described in Section 2.1. For a more comprehensive treatment of the subject, we refer the reader to (Browne et al., 2012).

Let the search tree $T$ be a DAG [1] consisting of a set of nodes $\mathcal{S}_T \subseteq \mathcal{S}_{\neg f}$ and edges $\mathcal{A}_T \subseteq \mathcal{A}$, where $\mathcal{S}_{\neg f} = \mathcal{S}/s_f$ and $\mathcal{A}_{\neg f} = \mathcal{A}/\{(x, s_f) : x \in \mathcal{X}\}$. Let $Q_T(s, s')$ be the search value function for edge $(s, s') \in \mathcal{A}_T$. The MCTS algorithm iteratively builds $T$ through repeated application of three construction steps: *Select*, *Expand*, and *Backup*.

The search tree is initialized without any information: $\mathcal{S}_T = \{s_0\}$, $\mathcal{A}_T = \{\}$. In the *Select* step, a new node $s' \notin \mathcal{S}_T$ is visited by sampling a trajectory through the search tree $(s_0 \to \cdots \to s \to s')$ where each action is selected using a tree policy $P_T(s'|s)$. In the *Expand* step, the edge values $Q_T(s', s'')$ from the new node $s'$ to each of its children $s'' \in Ch(s')$ are initialized using a heuristic approximation. Finally, in the *Backup* step information from the new state $s'$ is propagated back up along the path in reverse order, starting from $s$ and moving towards $s_0$.

---

[1] Despite the name, most implementations of MCTS construct a DAG, not a tree (Cazenave et al., 2012). However, in many applications the search DAG is sparse and tree-like.

Like many sampling/search methods, MCTS can be used for both optimization (i.e. finding $\arg\max_x R(x)$) and integration (i.e. calculating $\sum_x R(x)$). Details about each step in the algorithm, such as how the paths through the search tree are selected (i.e. greedy vs stochastic), how the value functions are initialized (i.e. Monte Carlo rollout vs neural network prediction) and how the value functions are updated in the Backup step, can vary depending on the specific use case. In the next section, we discuss a particular implementation of MCTS that can be used to solve the GFlowNet problem in a tree-structured environments.

## 2.3 MCTS FOR APPROXIMATE INFERENCE

Suppose $G$ is tree-structured environment with terminal energies. Previous work has shown that MCTS can be used to calculate the true distribution $P(x) \propto R(x)$: we briefly outline this work below.

First, observe that in tree-structured environments, each state $s$ can only be reached by a unique trajectory $\tau = (s_0, \ldots, s)$.

The reward function $R(s, s')$ is defined by equation 2:

$$R(s, s') = \begin{cases} -\mathcal{E}(s) & \text{if } s' = s_f \\ 0 & \text{otherwise} \end{cases} \tag{2}$$

Further, suppose that the Backup step on a single pair of nodes $(s, s')$ is given by Equation 3:

$$Q_T(s, s') \leftarrow R(s, s') + Q_T(s') \tag{3}$$

Note that $Q_T(s) = \log \sum_{s' \in Ch(s)} \exp Q_T(s, s')$ is simply the search value function for the node $s$. This equation is reminiscent of the Soft-Bellman backup equation used in maximum-entropy RL (Haarnoja et al., 2017).

If the tree is fully constructed, i.e. $\mathcal{S}_T = \mathcal{S}_{\neg f}$ and $\mathcal{A}_T = \mathcal{A}$, it is possible to show that the tree value functions are equivalent to the optimal maximum-entropy state-action value functions (Buesing et al., 2019). This is stated formally in Theorem 1:

**Theorem 1 (Search Tree Consistency)** *If the search tree $T$ is constructed exhaustively using Equation 3 for the Backup step, then $Q_T(s, s') = \sum_{x:s \prec x} R(x)$ for all $(s, s') \in \mathcal{A}_T$.*

Note that this result does not require a particular strategy for sampling search tree paths in the Select step, nor does it require a specific value function initialization in the Expand step. Theorem 1 can also be extended to environments with intermediate energies (see Buesing et al. 2019 for full details). While useful, this approach does not generalize to non-tree structured environments, such as the common GFlowNet benchmark Hypergrid (Bengio et al., 2021a). Briefly, if this Backup is applied to such an environment, the search tree values $Q_T(s, s')$ may be biased by the number of unique trajectories leading to each terminating state (see Figure 2); in tree-structured environments, this is not an issue since every state can only be reached by a single trajectory (refer to Bengio et al. 2021b for further details).

## 3 RELATED WORK

MCTS (Kocsis & Szepesvári, 2006; Kocsis et al., 2006) has long been used for planning in Markov Decision Processes Browne et al. (2012). Early approaches employed simple methods for estimating state values, such as performing Monte Carlo rollout with a heuristic policy. More recent approaches (Anthony et al., 2017; Silver et al., 2017) have replaced these heuristics with neural networks, which are faster and can in principle generalize to regions of the space that are unseen. Remarkably, it has been demonstrated that these networks can effectively be trained with the samples that they themselves generate through sampling, even in cases where the dynamics of the environment are unknown (Schrittwieser et al., 2019).

Maximum Entropy Reinforcement Learning (MaxEnt RL, Fox et al. (2015)) differs from standard RL by seeking a balance between maximizing rewards and maintaining diversity. This approach can improve exploration and make control more robust when the model is imperfect. It can also be viewed as a form of probabilistic inference, where the optimal policy samples trajectories proportion to their reward Levine (2018). More recently, MaxEnt RL has been linked to GFlowNets through reward shaping Tiapkin et al. (2024); Deleu et al. (2024); Mohammadpour et al. (2024). This provides a framework for converting existing maximum-entropy RL algorithms into equivalent GFlowNet algorithms: for example, the Detailed Balance GFlowNet algorithm (Bengio et al., 2021b) can be expressed as Soft Q-Learning (Haarnoja et al., 2017) with a particular type of reward shaping. MCTS can also be modified to work with MaxEnt RL (Xiao et al., 2019) and perform inference in tree-structured environments (Buesing et al., 2019).

Our work also shares some similarities with a concurrent work (Morozov et al., 2024): however, our contributions differ in several ways. Our work is more general in the sense that it is applicable to multiple GFlowNet parameterizations (DB, SubTB, and FM) and works with both terminating-reward and intermediate-reward environments, while theirs is limited to Soft Q-Learning (i.e. DB) in terminating-reward environments. Additionally, the manner in which their tree is constructed differs considerably from our own approach: theirs is similar to AlphaZero in the sense that it evolves several independent search trees simultaneously, while ours is more like TreeSample (Buesing et al., 2019) in that it builds one large search DAG (in parallel) from which trajectories can be sampled during training, somewhat like a replay buffer. Finally, the way they use search to influence training is distinct: their approach relies on taking the flow estimates from the search tree directly as training targets, while our approach only constructs the tree for the purposes of sampling.

## 4 METHODOLOGY

### 4.1 MONTE CARLO DAG SEARCH (MCDS)

In this section, we describe how to iteratively build a search DAG $D$ using MCTS-inspired graph operations. We call this approach Monte Carlo DAG Search (MCDS).

Like with the search tree $T$ in MCTS, the search DAG $D$ consists of a set of nodes $\mathcal{S}_D \subseteq \mathcal{S}_{\neg f}$ and edges $\mathcal{A}_D \subseteq \mathcal{A}$ that corresponds to a connected subgraph of the environment DAG $G$. Let $F_D(s, s')$ be the search DAG flow function for edge $(s, s')$, and let $R(s, s')$ be the reward function (defined below).

The particular manner in which the flow functions are updated for each action $(s, s')$ in the trajectory is described in Equation 4:

$$\log F_D(s, s') \leftarrow R(s, s') + \log F_D(s') \tag{4}$$

Note that $F_D(s) = \sum_{s' \in Ch(s)} F_D(s, s')$ is simply the state flow in the search DAG $D$. This equation is nearly identical to the Soft-Bellman backup in Equation 3, although the value functions have been replaced with flow functions.

Assuming that $G$ corresponds to a terminating reward environment, we can define the edge reward function as follows:

$$R(s, s') = \begin{cases} -\mathcal{E}(s) & \text{if } s' = s_f \\ \log P_B(s|s') & \text{otherwise} \end{cases} \tag{5}$$

This definition corresponds to the reward shaping for maximum entropy RL described in (Tiapkin et al., 2024; Deleu et al., 2024). If the DAG construction proceeds until the $D$ covers the entire environment graph $G$ (i.e. $\mathcal{S}_D = \mathcal{S}_{\neg f}$ and $\mathcal{A}_D = \mathcal{A}$), Theorem 2 asserts that the DAG flows are equal to the optimal GFlowNet flows (a proof can be found in Appendix 7.2).

**Theorem 2 (Search DAG consistency)** *If the search DAG $D$ is constructed exhaustively using Equation 4 for the Backup step, then $F_D(s, s') = F(s, s')$ for all $(s, s') \in \mathcal{A}$.*

A simple corollary is that the tree policy $P_D(s'|s) \propto F_D(s, s')$ can be used to sample terminating states $x \in \mathcal{X}$ proportional to $R(x)$. Let $P_D(x) = \sum_{\tau:(x, s_f) \in \tau} P_D(\tau)$.

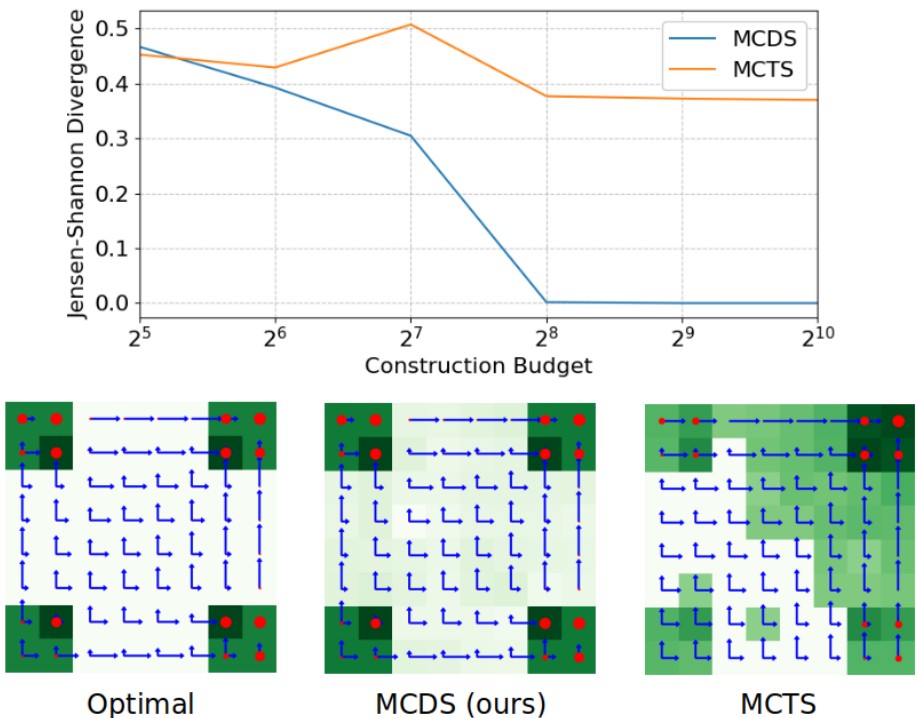

Figure 2: With sufficient budget, MCDS can approximate the true flows, forward policies, and terminating distributions in an 8x8 Hypergrid environment. Each square represents a state in the hypergrid, with $s_0 = (0,0)$ in the bottom left. The green shading represents the terminating distribution $P(s)$, and the arrows and dots represent the forward policy $P_F(s'|s)$, with the red dot indicating the terminating probability $P_F(s_f|s)$. The MCTS solution is biased towards states further from the origin since they can be reached with a larger number of unique trajectories.

**Corollary 2.1** *If $D$ is exhaustive, then the terminating state distribution $P_D(x)$ induced by the tree policy $P_D(s'|s)$ is equal to the true terminating state distribution $P(x) \propto R(x)$.*

Following (Deleu et al., 2024), the reward can be modified slightly to accommodate DAG construction in intermediate energy environments:

$$R(s, s') = \begin{cases} 0 & \text{if } s' = s_f \\ \mathcal{E}(s) - \mathcal{E}(s') + \log P_B(s|s') & \text{otherwise} \end{cases} \quad (6)$$

Theorem 2 can be extended to both intermediate-energy and terminating-energy environments; the proof in Appendix 7.2 covers both cases.

In terminating reward environments, it is possible to run MCDS without requiring additional reward function evaluations during construction. This can be accomplished by modifying Equation 5 such that $R(s, s') = F_D(s, s')$ when $s' = s_f$. In this case, the MCDS can be viewed as simply aggregating flow estimates across multiple states. After exhaustive construction the DAG flows $F_D(s, s')$ will correspond to the flows for a distribution $\hat{P}(x) \propto \hat{R}(x)$, where $\hat{R}(x)$ is the value used to initialize $F_D(x, s_f)$ in the Expand step for the terminating state $x$. In Section 5.1 we demonstrate that this approach can be useful for training.

### 4.2 APPLYING MCDS FOR GFLOWNET TRAINING

Most GFlowNet algorithms involve sampling trajectories from the environment and minimizing a differentiable loss on these samples with gradient descent. The precise form of the loss function depends on the particular GFlowNet parameterization and training objective. However, regardless

of parameterization, the sampling strategy is a critical part of the optimization can have a large impact on overall performance.

Let $P_M(s'|s)$ denote the sampling policy. In principle, the only requirement of the sampling policy is that it has full support over the set of trajectories $\mathcal{T}$. The most basic strategy therefore is to sample trajectories on-policy using the current model's parameters. In the case of DB, SubTB, and TB, the learned forward policy can be used $P_M(s'|s) = P_F(s'|s;\theta)$. In the case of FM, which does not parameterize a forward policy directly, the sampling policy can be defined using the edge flow function: $P_M(s'|s) \propto F(s, s';\theta)$.

Our method involves constructing a search DAG $D$ with MCDS and drawing samples with $P_M(s'|s) = P_D(s'|s)$. Inspired by previous works combining MCTS with RL (Silver et al., 2017; Buesing et al., 2019; Xiao et al., 2019), we can guide construction of $D$ by using the current GFlowNet flow estimates in the Expand step. First, let us consider the DB objective (Bengio et al., 2021b), which requires parameterizing a forward policy $P_F(s'|s;\theta)$, a state flow function $F(s;\theta)$, and (optionally) a backward policy $P_B(s|s';\theta)$. In this case we can apply the flow identity $F(s, s';\theta) = F(s;\theta)P_F(s'|s;\theta)$ and initialize tree flows for new nodes using Equation 7:

$$\log F_D(s, s') \leftarrow \log F(s, s';\theta) \tag{7}$$

This approach also works for the SubTB objective (Madan et al., 2023), since it parameterizes the distribution in the same manner. For the FM (Bengio et al., 2021a) case, we can use the learned state-action flow $F(s, s';\theta)$ directly.

In the intermediate reward case the forward-looking flow $\tilde{F}(s, s';\theta)$ (Pan et al., 2023) is used in combination with the intermediate energy $\mathcal{E}(s)$, as described in Equation 8 (see Appendix 7.2 for justification):

$$\log F_D(s, s') \leftarrow \log \tilde{F}(s, s';\theta) - \mathcal{E}(s) \tag{8}$$

As tree construction progresses, the tree flows $F_D(s, s')$ move away from the GFlowNet estimates $F(s, s';\theta)$ and towards the optimal flows $F(s, s')$. Exhaustive tree construction is usually impractical; in cases where it is feasible, learning an approximation $P_F(s'|s;\theta)$ is superfluous since the DAG distribution $P_D(x)$ perfectly models the distribution over terminating states $P(x)$. In practice, we build $D$ stochastically using a fixed budget $B \ll |\mathcal{A}|$ of construction iterations. The method for sampling from the (usually incomplete) search DAG is described in Equation 9:

$$P_M(s'|s) = \begin{cases} P_D(s'|s) & \text{if } s \in \mathcal{S}_D \\ P_F(s'|s;\theta) & \text{otherwise} \end{cases} \tag{9}$$

Empirically we find that mixing samples from $P_M(s'|s)$ and $P_F(s'|s;\theta)$ in a 1:1 ratio produces the best results. We can observe that sampling trajectories from the optimal distribution $P_F(s'|s)$ does not necessarily lead to superior optimization: empirically, simple on-policy training can result in faster convergence under certain conditions (Atanackovic & Bengio, 2024). Intuitively, it is important for the sampling policy to capture regions of the space where the current model and the optimal distribution differ. Focusing exclusively on the modes of the distribution might not be the best strategy for finding such states.

Building $D$ every iteration can be quite inefficient, since the GFlowNet policy $P_F(s'|s;\theta)$ does not change much after a single gradient update. It also slows down training dramatically, since each time $D$ is constructed the model $F(s'|s;\theta)$ and energy function $\mathcal{E}(s)$ are queried several times. In our experiments $D$ is built every few iterations, and the construction operations are executed in parallel (see Algorithm 1 for full details).

## 5 EXPERIMENTS

### 5.1 HYPERGRID

First we evaluate our method on the standard Hypergrid GFlowNet benchmark from (Bengio et al., 2021a). Hypergrid is a $D$-dimensional grid environment of size $H^D$ where every state is terminating. It uses a sparse, multi-modal reward function that is concentrated near each of the $2^D$ corners of the

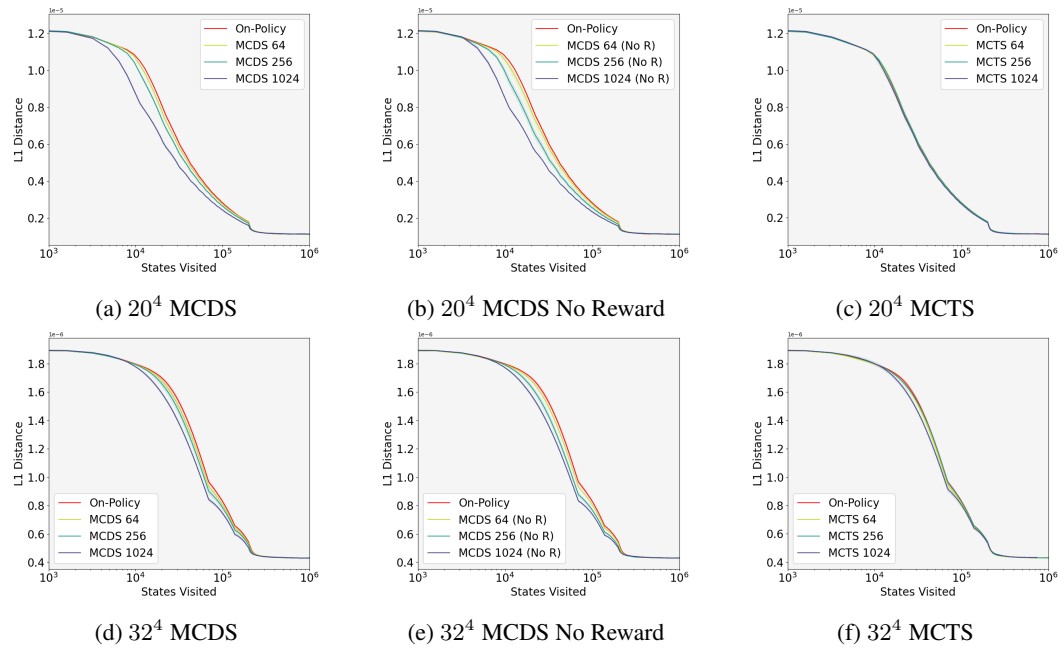

Figure 3: Hypergrid experiments with varying grid sizes and MCDS configurations. MCDS works both with and without access to reward function during tree construction. Larger tree budgets help with MCDS, but are not helpful with MCTS. Experiments run with 3 seeds, mean and standard deviation reported. The reported metric is average L1 distance $|P(x) - P(x;\theta)|$ over states $x \in \mathcal{X}$.

hypergrid. The initial state is the origin $(0)^D$ located at the corner of the hypergrid. Each action is a step that increments one of the $D$ coordinates by 1 (up to a maximum of $H - 1$).

For our experiments we use the sparser formulation of the hypergrid reward ($R_0 = 0.0001, R_1 = 1.0, R_2 = 3.0$), and focus on two large environments ($D = 4, H \in \{20, 32\}$). We compare on-policy training using the DB objective (Bengio et al., 2021b) with different configurations of MCDS and MCTS. The results are summarized in Figure 3. MCDS (Figures 3a and 3d) improves training compared to on-policy sampling, with larger tree construction budgets providing a bigger improvement. Furthermore, we show that MCTS (Figures 3c and 3f) does not meaningfully improve training with equal construction budgets, and may even harm it. We also show how variants of MCDS that do not query the reward function during construction (Figures 3b and 3e) can improve convergence.

## 5.2 BLOCKSWORLD

### 5.2.1 TASK DESCRIPTION

We have done extensive experiments with the Blocksworld (Valmeekam et al., 2023) planning problems to test our methodology in a language model reasoning task. In this task, the model is required to produce a sequence of actions to rearrange blocks into stacks in a specified order. A state $s$ represents the current arrangement of the blocks, and each action is a written instruction for moving the blocks. The actions use one of four verbs—STACK, UNSTACK, PUT, or PICKUP—along with the corresponding objects. We generate valid actions based on domain constraints and the current block configuration, and query the language model to estimate the flow $F(s;\theta)$ and forward policy $P_F(s'|s;\theta)$. Based on the current state and the action taken, the next state can be obtained in a deterministic fashion. The planning process terminates when the maximum number of steps is reached, such that all trajectories have the same length. A step count is used to prevent cycles and enforce the DAG structure of the environment. The reward for a terminating state $x$ is a function of how well the current block configuration meets the goal criteria specified in the environment definition. Let $f(x)$ be the fraction of the criteria satisfied in state $x$; if $f(x) = 1$ then $R(x) = 100$, otherwise

Table 1: Results on the Blocksworld task with different difficulty levels, with the number of test examples (environments) indicated in brackets. Acc = accuracy @ 20, Reward = average reward @ 20. Mean and standard deviations reported over five seeds.

| Method | 2-step (15) | | 4-step (42) | | 6-step (99) | | 8-step (138) | |
|---|---|---|---|---|---|---|---|---|
| | Acc (%) | Reward | Acc (%) | Reward | Acc (%) | Reward | Acc (%) | Reward |
| CoT (2-shot) | $37.3_{\pm 8.9}$ | $5.6_{\pm 0.8}$ | $6.7_{\pm 5.2}$ | $1.0_{\pm 0.0}$ | $3.0_{\pm 1.6}$ | $0.4_{\pm 0.1}$ | $1.3_{\pm 0.6}$ | $0.7_{\pm 0.0}$ |
| CoT (5-shot) | $40.1_{\pm 14.1}$ | $5.0_{\pm 1.0}$ | $4.8_{\pm 2.7}$ | $0.9_{\pm 0.1}$ | $3.0_{\pm 0.0}$ | $0.5_{\pm 0.0}$ | $2.3_{\pm 0.0}$ | $0.5_{\pm 0.1}$ |
| DB | | | | | | | | |
|    On-Policy | $81.3_{\pm 21.8}$ | $41.7_{\pm 20.8}$ | $80.0_{\pm 10.0}$ | $17.4_{\pm 2.5}$ | $41.8_{\pm 20.6}$ | $4.5_{\pm 2.0}$ | $6.7_{\pm 2}$ | $1.8_{\pm 0.2}$ |
|    MCDS | $96.0_{\pm 6.0}$ | $69.1_{\pm 7.2}$ | $81.4_{\pm 4.6}$ | $31.3_{\pm 15.0}$ | $73.7_{\pm 7.3}$ | $23.1_{\pm 5.6}$ | $20.3_{\pm 7.6}$ | $2.4_{\pm 1.1}$ |
| SubTB | | | | | | | | |
|    On-Policy | $90.7_{\pm 10.1}$ | $74.5_{\pm 8.4}$ | $50.5_{\pm 21.2}$ | $22.4_{\pm 12.9}$ | $37.8_{\pm 21.6}$ | $8.1_{\pm 5.5}$ | $7.3_{\pm 3.4}$ | $2.4_{\pm 0.3}$ |
|    MCDS | $90.7_{\pm 10.1}$ | $78.7_{\pm 9.7}$ | $73.3_{\pm 12.4}$ | $36.4_{\pm 12.5}$ | $68.1_{\pm 7.4}$ | $23.1_{\pm 5.6}$ | $38.4_{\pm 11.6}$ | $5.1_{\pm 1.3}$ |
| TB | $86.7_{\pm 13.3}$ | $75.6_{\pm 11.7}$ | $57.1_{\pm 15.1}$ | $28.7_{\pm 10.6}$ | $32.5_{\pm 24.5}$ | $10.9_{\pm 12.5}$ | $4.1_{\pm 2.3}$ | $2.2_{\pm 0.7}$ |
| TBVar | $94.7_{\pm 5.6}$ | $81.5_{\pm 11.0}$ | $39.5_{\pm 15.0}$ | $13.8_{\pm 7.4}$ | $34.5_{\pm 25.0}$ | $9.0_{\pm 6.6}$ | $3.3_{\pm 1.2}$ | $1.7_{\pm 0.7}$ |

$R(x) = 10f(x)$. For example, it could be the case that in the initial state, the orange block is on the table, the blue block is on the table and the hand is empty. A valid action in this case would be to pickup the orange block. The goal criteria of the environment could be that the orange block ends up on top of the blue block.

### 5.2.2 TRAINING SETUP

The maximum number of steps needed to reach the goal from the initial state defines the task's difficulty. The distribution of tasks is as follows: 30 examples require 2 steps, 57 examples require 4 steps, 114 examples require 6 steps, and 153 examples require 8 steps. Based on the setup from (Hao et al., 2023), we choose the first 15 examples from each group as training, with the remaining ones used as test samples. We show the accuracy and average reward of different methods for these groups in the table 1. During the test phase, for each environment (example) we sample 20 trajectories and if any trajectory reaches the goal, we consider the instance solved. All experiments are done with 5 random seeds and the mean and standard deviation are reported. Further details about the Blocksworld task and training can be found in Appendix 7.3

### 5.2.3 RESULTS

In all experiments, we fine-tuned the LLama3 8B model (Dubey et al., 2024) to predict policies and flows. The base model, without fine-tuning, was unable to produce admissible results in any of the evaluated settings. However, fine-tuning the model using any of the baseline GFN methods consistently resulted in improved performance. Notably, incorporating MCTS significantly enhanced GFN training across all configurations. Furthermore, as task difficulty increased, the performance gap widened, emphasizing the impact of MCTS in this challenging reasoning experiment. In Table 1, TB corresponds most closely to (Hu et al., 2023), while TBVAR aligns with the approach in (Yu et al., 2024), which uses the modified TB objective from (Zhang et al., 2023).

### 5.3 FACTOR GRAPHS

The *Factor Graphs* benchmark, originally proposed in (Buesing et al., 2019) but reformulated for GFlowNets in (Deleu et al., 2024), is a challenging discrete inference task. Each Factor Graph environment corresponds to a factorizable distribution over $N$ categorical variables (each with support size $K$). Notably, since each factor only depends on a subset of the $N$ variables, intermediate rewards can be given once those variables have been assigned. Each action in the environment corresponds to the assignment of one of the $N$ variables, resulting in a total of $(K+1)^N$ states of which $K^N$ are terminating.

We consider two environments: the Permuted Chain environment and the Factorgraphs1 environment (see Section 7.3 for more details). As described in Section 4.1, MCDS construction can proceed with or without intermediate rewards. The forward-looking (FL) variants that use intermediate rewards for both tree construction and loss calculation clearly outperform those that do not, as shown

in Figures 4a and 4c. Furthermore, MCDS does seem to improve over on-policy in the FL case, although in the terminating reward case MCDS and on-policy sampling both perform poorly. Figures 4b and 4d demonstrate that MCDS also works with SubTB FL, although both SubTB FL methods seem to be more unstable than their DB FL counterparts.

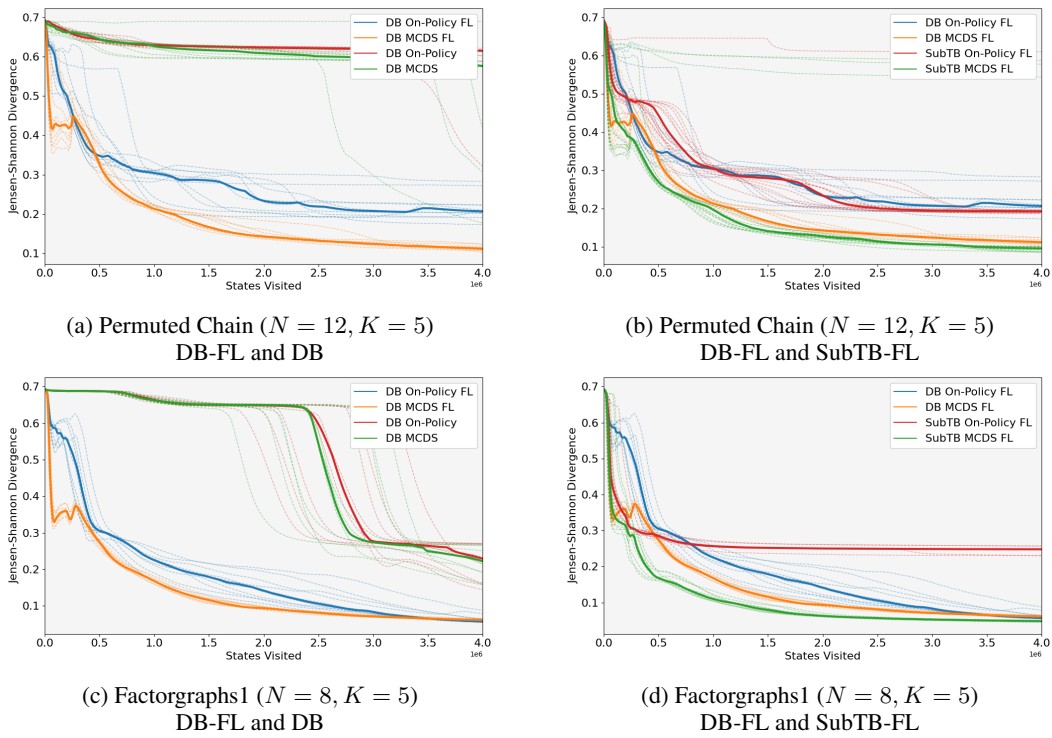

(a) Permuted Chain ($N = 12, K = 5$)
DB-FL and DB

(b) Permuted Chain ($N = 12, K = 5$)
DB-FL and SubTB-FL

(c) Factorgraphs1 ($N = 8, K = 5$)
DB-FL and DB

(d) Factorgraphs1 ($N = 8, K = 5$)
DB-FL and SubTB-FL

Figure 4: Experiments in two different Factorgraphs environments (Permuted Chain and Factorgraphs1) with different GFN objectives (DB and SubTB). The thin dashed lines represent individual trajectories for 10 seeds; the thick lines represent the median across seeds. MCDS consistently results in faster convergence in the forward-looking (FL) case.

## 6  CONCLUSION

In this work, we propose Monte Carlo DAG Search (MCDS), a novel adaptation of MCTS to the GFlowNet problem. Our method employs reward shaping to modify the Backup step in maximum entropy MCTS so that it can apply to GFlowNets. We show that our approach can be used to calculate optimal flows in both terminating and intermediate reward environments. By employing MCDS as a tool for sampling the environment, we demonstrate how it can improve GFlowNet training. Through a series of experiments covering different state spaces, reward structures, neural network architectures, and GFlowNet parameterizations, we demonstrate the broad applicability and effectiveness of our method for GFlowNet training.

There are several promising directions for future work. Our current MCDS formulation requires parameterizing a state flow $F(s)$ or state-action flow $F(s, s')$, which limits its applicability to the DB, SubTB, and FM parameterizations. However, it may be possible to develop a strategy that works with TB. Furthermore, we have not explored combining MCDS with other successful GFlowNet sampling methods like replay buffers, local search (Kim et al., 2024), and Thompson sampling (Rector-Brooks et al., 2023), which could further improve performance. Finally, it would be valuable to explore different formulations of the DAG policy $P_D(s'|s)$ that is used in tree construction. Our approach is most similar to MENTS (Xiao et al., 2019), but it may be possible to consider other variants (Buesing et al., 2019) which offer a different balance of exploration and exploitation.

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

# 7 APPENDIX

## 7.1 MCDS FULL ALGORITHMS

---

**Algorithm 1** Monte Carlo DAG Search

---

**Require:** Environment graph $G$, energy function $\mathcal{E}(s)$, flow function $F(s, s'; \theta)$, backward policy
$P_B(s|s'; \theta)$, budget $B$, worker count $W$
1: Initialize $\mathcal{S}_D = \{s_0\}, \mathcal{A}_D = \{\}, b = 0$
2: **while** $b < B$ **do**
3:     $w \leftarrow \min(W, B - b)$
4:     **for** $i \in \{0, \cdots w - 1\}$ **do**
5:         $s_i \leftarrow \emptyset, s_i' \leftarrow s_0, \tau_i \leftarrow ()$                 ▷ Start SELECT step
6:         **while** $s_i' \in S_D$ and $s_i' \neq s_f$ **do**
7:             $s \leftarrow s_i'$
8:             $C \leftarrow \{s' \in Ch(s) : (s, s') \notin \mathcal{A}_D\}$
9:             $P_D(s'|s) \leftarrow F_D(s, s') / \sum_{s'' \in C} F_D(s, s'')$ for all $s' \in C$
10:            $s' \sim P_D(s'|s)$
11:            $s_i' \leftarrow s', s_i \leftarrow s, \tau_i \leftarrow \tau_i \cdot (s, s')$
12:        **end while**
13:        **if** $s_i' \neq s_f$ **then**                     ▷ Start EXPAND step
14:            $F_D(s_i', s'') \leftarrow F(s_i', s''; \theta)$
15:        **end if**
16:        $\mathcal{S}_D \leftarrow \mathcal{S}_D \cup \{s_i'\}$
17:        $\mathcal{A}_D \leftarrow \mathcal{A}_D \cup \{(s_i, s_i')\}$
18:        **for** $j \in \{|\tau_i| - 1, \cdots, 0\}$ **do**           ▷ Start BACKUP step
19:            $(s, s') \leftarrow \tau_i[j]$
20:            **if** $s' = s_f$ **then**
21:                $F_D(s') \leftarrow 0$
22:            **else**
23:                $F_D(s') \leftarrow \sum_{s'' \in Ch(s')} F(s', s'')$
24:            **end if**
25:            $\log F_D(s, s') \leftarrow R(s, s') + \log F(s')$
26:        **end for**
27:     **end for**
28:     $b = b + w$
29: **end while**
30: **return** $D, F_D(s, s')$

---

## 7.2 PROOF OF MCDS DAG CONSISTENCY

We define MCDS (Algorithm 1, using Backup Equation 4)) as being run to completion if $\mathcal{S}_D = \mathcal{S}_{\neg f}$
and $\mathcal{A}_D = \mathcal{A}$.

Let $G$ be an environment with associated reward $R(x)$. First, we will prove the terminating-reward
case, i.e. $R(s) = 0$ for all $s \notin \mathcal{X}$.

**Claim 1** *In a terminating reward environment, if MCDS is run to completion, then $F_D(s, s') = F(s, s')$ for all $(s, s') \in \mathcal{A}$, where $F(s, s')$ is the optimal edge flow induced by the environment $G$ and the reward function $R(s)$.*

**Proof 1** *Let $L(s)$ be the length of the longest trajectory from $s$ to $s_f$, using edges in $\mathcal{A}$.*

*Let $N = \max_{s \in \mathcal{S}_{\neg f}} L(s)$.*

*We will prove the claim by induction on $L(s)$.*

***Base case:*** *Assume $L(s) = 0$*

*If $L(s) = 0$, then $s \in \mathcal{X}$ by definition.*

---

**Algorithm 2** MCDS GFlowNet Training

---

**Require:** Environment graph $G$, energy function $\mathcal{E}(s)$, budget $B$, worker count $W$, training itera-
   tions $I$, batch size $J$, build frequency $K$, loss function $\mathcal{L}$
 1: Initialize $\theta$
 2: **for** $i \in \{0, \cdots, I-1\}$ **do**
 3:    **if** $i \mod K = 0$ **then**
 4:        $D, F_D(s, s') \leftarrow \text{MCDS}(G, \mathcal{E}, B, W)$
 5:    **end if**
 6:    $S = \{\}$
 7:    **for** $j \in \{0, \cdots, J-1\}$ **do**
 8:        $s_j \leftarrow s_0$
 9:        **while** $s_j \neq s_f$ **do**
10:            $s \leftarrow s_j$
11:            **if** $s \in \mathcal{S}_D$ **then**
12:                $P_B(s'|s) \leftarrow F_D(s, s')/\sum_{s'' \in Ch(s)} F_D(s, s'')$
13:                $s' \sim P_B(s'|s)$
14:            **else**
15:                $s' \sim P(s'|s; \theta)$
16:            **end if**
17:            $s_j \leftarrow s'$
18:        **end while**
19:    **end for**
20:    $l = \frac{1}{|S|}\mathcal{L}(S, \theta)$
21:    $\theta \leftarrow \theta + \nabla_\theta l$
22: **end for**
23: Return $\theta$

---

*In this case, $F_D(s, s_f) = R(s, s_f) = R(s)$ by the Backup equation.*

*Terminating reward environments have the property that $F(s, s_f) = R(s)$, thus $F_D(s, s_f) = F(s, s_f)$.*

***Inductive case:*** *Assume the claim holds for $L(s) < n$, we want to prove it for $L(s) = n$.*

*If $L(s) = n$, then by definition each node $s' \in Ch(s)$ has $L(s') < n$.*

*By the inductive hypothesis, $F_D(s', s'') = F(s', s'')$ for all $s'' \in Ch(s')$.*

*This implies $F_D(s') = \sum_{s'' \in Ch(s')} F_D(s', s'') = F(s')$.*

*By the Backup equation,*

$$
\begin{aligned}
\log F_D(s, s') &= R(s, s') + \log F_D(s') \\
&= \log P_B(s|s') + \log F(s') \\
&= \log F(s, s')
\end{aligned}
$$

*Therefore, $F_D(s, s') = F(s, s')$ for $L(s) = n$, completing the induction.*

∎

Now, we will prove the intermediate reward case, again by using induction on $L(s)$.

**Claim 2** *In an intermediate reward environment, if MCDS is run to completion, then $F_D(s, s') = \tilde{F}(s, s')$ for all $(s, s') \in \mathcal{A}$, where $\tilde{F}(s, s')$ is the optimal forward-looking edge flow induced by the environment $G$ and the reward function $R(s)$.*

**Proof 2** *Base case: Assume $L(s) = 0$*

*In this case, $F_D(s, s_f) = R(s, s_f) = 0$ by the Backup equation.*

*Intermediate reward environments have the property that $\tilde{F}(s, s_f) = 0$, thus $F_D(s, s_f) = \tilde{F}(s, s_f)$.*

*Inductive case: Assume the claim holds for $L(s) < n$, want to prove $L(s) = n$.*

*By the inductive hypothesis, $F_D(s', s'') = \tilde{F}(s', s'')$ for all $s'' \in Ch(s')$.*

*This implies $F_D(s') = \sum_{s'' \in Ch(s')} F_D(s', s'') = \tilde{F}(s')$.*

*By the Backup equation,*

$$
\begin{aligned}
\log F_D(s, s') &= R(s, s') + \log F_D(s') \\
&= \mathcal{E}(s) - \mathcal{E}(s') + \log P_B(s|s') + \log \tilde{F}(s') \\
&= \log F(s) + \log P_B(s|s') - \mathcal{E}(s') \\
&= \log F(s, s') - \mathcal{E}(s') \\
&= \log \tilde{F}(s, s')
\end{aligned}
$$

*Therefore, $F_D(s, s') = \tilde{F}(s, s')$ for $L(s) = n$, completing the induction.*

∎

### 7.3 EXPERIMENTAL DETAILS

Here we provide more details about the training and the benchmarks.

#### 7.3.1 HYPERGRID

The hypergrid reward takes the form described in Equation 10, where $H \in \mathbb{N}$ is the height of the grid, $D \in \mathbb{N}$ is the dimension, and $R_0, R_1, R_3 \in \mathbb{R}^+$ are parameters that control sparsity. Each hypergrid environment has $|\mathcal{X}| = H^D$, $|\mathcal{S}| = H^D + 1$, and $|\mathcal{A}| = D(H^D - H^{D-1})$.

$$
R(x) = R_0 + R_1 \prod_{d=1}^{D} \mathbb{I}\left[ 0.25 < \left| \frac{x_d}{H-1} - 0.5 \right| \right] + R_2 \prod_{d=1}^{D} \mathbb{I}\left[ 0.3 < \left| \frac{x_d}{H-1} - 0.5 \right| < 0.4 \right] \quad (10)
$$

Following previous work (Madan et al., 2023), we use a simple 2-layer 256-dimensional MLP with weight typing to parameterize the flow and policy functions $F(s; \theta)$, $P_F(s'|s; \theta)$ and $P_B(s|s'; \theta)$. We do not employ $\epsilon$-uniform exploration or replay buffers for any of the methods. We run experiments with a batch size of 16 for 62500 steps, resulting in 1 million sampled trajectories. The learning rate is set to 1e-3. Training statistics are calculated using a moving average of the last 200,000 trajectories sampled on-policy from the model.

#### 7.3.2 BLOCKSWORLD

In all the experiments, we finetune Llama3 8B with LoRA (Hu et al., 2021) with $r = 32$, $\alpha = 64$, and dropout=0.1. The learning rate is set to 2e-5 and the number of trajectories is set to 20. Since the study is about investigating the effect of MCTS on GFlowNet methods, we avoid learning rate, reward, and sampling temperature scheduling. For all methods we use a uniform backwards policy and do not employ $\epsilon$-uniform exploration or replay buffers.

An example prompt for a 4-step example is given in Table 2.

The prompt format and instructions do not vary across tasks or states, but the goal, in-context examples, and current state information do. In Table 2, <current state> and <goals> are filled with the corresponding status of the current state and task goal.

The sizes of each of the environments and the MCDS budgets used for each experiment are summarized in Table 3.

I am playing with a set of blocks where I need to arrange the blocks into stacks.
Here are the actions I can do:
Pick up a block
Unstack a block from on top of another block
Put down a block
Stack a block on top of another block
I have the following restrictions on my actions:
I can only pick up or unstack one block at a time.
I can only pick up or unstack a block if my hand is empty.
I can only pick up a block if the block is on the table and the block is clear.
A block is clear if the block has no other blocks on top of it and if the block is not picked up.
I can only unstack a block from on top of another block if the block
I am unstacking was really on top of the other block.
I can only unstack a block from on top of another block if the block I am unstacking is clear.
Once I pick up or unstack a block, I am holding the block.
I can only put down a block that I am holding.
I can only stack a block on top of another block if I am holding the block being stacked.
I can only stack a block on top of another block if the block onto which I am stacking the
block is clear.
Once I put down or stack a block, my hand becomes empty.
[STATEMENT]
As initial conditions I have that, the red block is clear, the blue block is clear, the yellow block is clear,
the hand is empty, the blue block is on top of the orange block, the red block is on the table, the orange
block is on the table, and the yellow block is on the table. My goal is to have that the orange block is on
top of the blue block.
My plan is as follows:
[PLAN]
unstack the blue block from on top of the orange block
put down the blue block
pick up the orange block
stack the orange block on top of the blue block
[PLAN END]
[STATEMENT]
As initial conditions I have that, <current state>
My goal is to have that <goals>
My plan is as follows:
[PLAN]
<action>

Table 2: 4-step prompt example

| Environment | $B$ | $|\mathcal{S}|$ | $|\mathcal{X}|$ | $|\mathcal{A}|$ |
|---|---|---|---|---|
| 2-step | 16 | 9 (8-13) | 5 (4-8) | 13 (11-21) |
| 4-step | 32 | 56 (11-136) | 29 (4-81) | 107 (15-283) |
| 6-step | 64 | 77 (27-522) | 37 (11-249) | 148 (45-1173) |
| 8-step | 100 | 58 (32-423) | 167 (75-1232) | 345 (142-2891) |

Table 3: BlocksWorld environment sizes (in terms of states $\mathcal{S}$, terminating states $\mathcal{X}$, and edges/transitions $\mathcal{A}$) and associated MCDS budgets $B$. Environment sizes are reported as median (min-max).

### 7.3.3 FACTOR GRAPHS

Our environments were constructed in the same manner as Deleu et al. (2024). However, we adjusted the parameters to create sparser environments with lower entropies. For the Permuted Chain environment we set the `rbf_scale` parameter to 2.5 and the `factor` parameter to 2.0, resulting in an entropy of approximately 3.97 (using the natural logarithm): for comparison, the uniform distribution has entropy of 12.42. For the Factorgraphs1 environment we set the `scale` parameter to 3.0, resulting in an entropy of approximately 2.84, compared with the uniform entropy of 12.88.

Each factor graph environment has $|\mathcal{S}| = 1 + (K + 1)^N$, $|\mathcal{X}| = K^N$, and $|\mathcal{A}| = K^N + \sum_{n=1}^{N-1} \binom{N}{n}(N - n)K^{n+1}$. The Permuted Chain environment ($K = 5$, $N = 12$) has $|\mathcal{S}| \approx 2e9$,

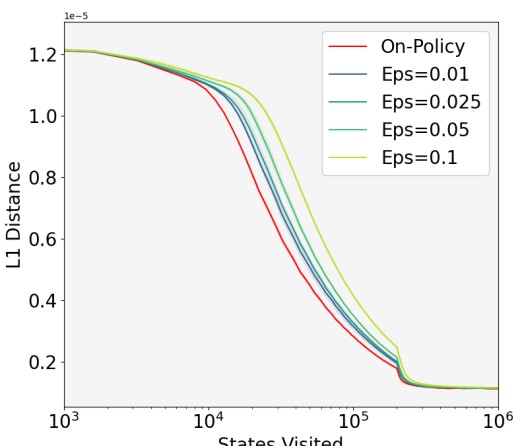

Figure 5: Hypergrid experiment testing $\epsilon$-uniform exploration.

$|\mathcal{X}| \approx$ 2e8, $|\mathcal{A}| \approx$ 2e10. The Factorgraphs1 environment ($K = 5$, $N = 8$) has $|\mathcal{S}| \approx$ 2e6, $|\mathcal{X}| \approx$ 4e5, $|\mathcal{A}| \approx$ 1e7.

Similar to our setup with Hypergrid, we use a simple 2-layer 256-dimensional MLP with weight typing to parameterize the flow and policy functions $F(s; \theta)$, $P_F(s'|s; \theta)$; the backward policy is uniform. We do not employ $\epsilon$-uniform exploration or replay buffers for any of the methods. We run experiments with a batch size of 128 for 62500 steps, resulting in 4 million sampled trajectories. The learning rate is set to 1e-4. Training statistics are calculated using a moving average of the last 200,000 trajectories sampled on-policy from the model.

### 7.4 EPSILON-UNIFORM EXPLORATION EXPERIMENTS

In the $20^4$ sparse hypergrid, on-policy sampling outperforms configurations with $\epsilon \in \{0.01, 0.025, 0.05, 0.1\}$, as demonstrated in Figure 5. Since MCDS outperforms on-policy training in this setting (Figure 3), it also outperforms the configurations with exploration.

### 7.5 RUNTIME COMPARISON

Constructing the MCDS DAG requires additional computation that can slow down training when compared to on-policy sampling. However, the magnitude of the slowdown depends on the construction budget $B$, the number of parallel workers $W$, and the build frequency $K$. Table 4 summarizes the relative slowdown of different MCDS variants used in the Hypergrid and Factor Graph experiments (DB parameterization). Note that the reported metrics include time associated with the calculation of rewards, losses, gradients, and evaluation metrics. With the configurations we tested, the total MCDS runtime penalty ranges from a factor of 1.30 to 3.47.

| Environment | $B$ | $W$ | $K$ | Ratio |
|---|---|---|---|---|
| $20^4$ Hypergrid | 64 | 16 | 1 | 1.55 |
| $20^4$ Hypergrid | 256 | 16 | 4 | 2.11 |
| $20^4$ Hypergrid | 1024 | 16 | 16 | 3.47 |
| $32^4$ Hypergrid | 64 | 16 | 1 | 1.30 |
| $32^4$ Hypergrid | 256 | 16 | 4 | 1.58 |
| $32^4$ Hypergrid | 1024 | 16 | 16 | 1.94 |
| $5^8$ Factorgraphs1 | 1024 | 16 | 16 | 1.75 |
| $5^{12}$ Permuted Chain | 1024 | 16 | 16 | 2.45 |

Table 4: Total runtime of different MCDS variants, relative to comparable on-policy variants. $B$ is budget, $W$ is worker count, $K$ is build frequency.

