# OpenReview forum: "Adapting Monte Carlo Tree Search for Generative Flow Network Training"
_ICLR.cc/2025/Conference — Submitted to ICLR 2025_

### Official Review · Reviewer_F11R · 2024-11-02

**Soundness:** 2
**Presentation:** 3
**Contribution:** 2
**Rating:** 5
**Confidence:** 4

**Summary:**

Authors propose Monte Carlo DAG Search (MCDS) — MCTS-like approach for planning in GFlowNets that can improve training efficiency. It stores a subgraph of the DAG environment with flow estimates for each edge. It is built iteratively and flow estimates are updated using their relation to the GFlowNet rewards and backward policy. A combination of the GFlowNet forward policy and the policy induced by MCDS flow estimates is used as a sampling strategy (denoted by $P_M$) to obtain trajectories for GFlowNet training. Authors claim it to offer improvements over on-policy training, and support their claims by providing experimental evaluation on three environments.

**Strengths:**

The paper is well-written and easy to follow. I had no problem understanding the motivation, the method and the experiments (apart from some specific details, see Weaknesses).

GFlowNets have found success in various areas over the years, but the strategies for sampling trajectories for their training have not been extensively explored and studied yet. MCDS is a creative and promising approach that falls in this category.

While the idea of employing MCTS to improve GFlowNets has already been explored in the literature, the proposed approach has a number of crucial differences from the method of [1], as outlined by the authors, so I believe it to be a valid contribution.

References:

[1] Nikita Morozov, Daniil Tiapkin, Sergey Samsonov, Alexey Naumov, Dmitry Vetrov. Improving GFlowNets with Monte Carlo Tree Search. ICML 2024 Workshop on Structured Probabilistic Inference & Generative Modeling

**Weaknesses:**

My main concerns lie within the soundness of the experimental evaluation.

Firstly, out of 3 tasks used for evaluation, 2 (hypergrids and factor graphs) environments are toy. By this I mean that the spaces of objects we work with are small enough  ($32^4$ for hypergrid and $5^{12}$ for factor graphs) that it is possible to iterate over all objects in reasonable time, thus the task of training a GFlowNet policy for sampling is artificial. Such environments can offer valuable insight into the performance of a method, but I still strongly advise the authors to include larger environments for the evaluation that are closer to real-world applications. A good example would be molecule generation task introduced in the seminal work [2], which is used as a standard benchmark in many GFlowNet papers.

Secondly, since the authors position the purpose of their approach to be improving trajectory sampling for GFlowNet training, I would like to see some baselines other than the standard on-policy training. E.g. replay buffer approaches [3, 4], and using a mixture of a forward policy and a uniform distribution [5], which were demonstrated to also improve the efficiency of GFlowNet training. As stated by the authors, combining MCDS with such methods could lead to further improvements, but putting some direct comparison to them is also crucial for understanding the efficiency of MCDS.

In addition, I have a concern over which policy is used for evaluation. A standard approach to calculate L1 metric on hypergrids is to use an empirical distribution of terminal states of the last $N$ trajectories sampled for training, and then compare it to the true reward distribution. If this approach is also used here, then the policy being evaluated would be not the GFlowNet forward policy $P_F$, but the combination policy $P_M$. Then, such evaluation would show the improvements coming not only from introducing a different distribution over trajectories for GFlowNet training (motivation stated by the authors), but also from the adjustment to the forward policy introduced by MCDS. Can the authors please provide more details on how the metric is calculated? For factor graph environment I have the same question.

A positive note here is that I believe that MCDS could generally be used during inference of a trained model to introduce an adjustment to the forward policy (by using $P_M$ instead of $P_F$ during inference), potentially leading to better approximation of the reward distribution. Some experiments on this could further strengthen the paper.

It will also be good to include some runtime measurements to see how much of a computational overhead MCDS adds in comparison to standard on-policy training.

A minor remark, if $Q(s, s')$ in Equation 3 is an entropy-regularized action value function defined consistently with the standard RL notation, the correct way to write it would be $Q(s, s') = R(s, s') + \log \sum_{s''} \exp Q(s', s'')$. On the other hand, if it is defined as an exponent of action value function, I believe there should be no logarithm in the equation on line 187.

There is a typo "benchmarkpr" on line 57.

The paper presents a very interesting and promising approach, but the current quality of experimental evaluation is not enough to recommend acceptance to a conference of this level. However, I believe that with improved and extended experimental evaluation, this work would be a solid contribution to some future venue.

References:

[2] Emmanuel Bengio, Moksh Jain, Maksym Korablyov, Doina Precup, Yoshua Bengio. Flow Network based Generative Models for Non-Iterative Diverse Candidate Generation. NeurIPS 2021

[3] Max W. Shen, Emmanuel Bengio, Ehsan Hajiramezanali, Andreas Loukas, Kyunghyun Cho, Tommaso Biancalani. Towards Understanding and Improving GFlowNet Training. ICML 2023

[4] Nikhil Vemgal, Elaine Lau, Doina Precup. An Empirical Study of the Effectiveness of Using a Replay Buffer on Mode Discovery in GFlowNets. ICML 2023 Workshop on Structured Probabilistic Inference & Generative Modeling

[5] Nikolay Malkin, Salem Lahlou, Tristan Deleu, Xu Ji, Edward Hu, Katie Everett, Dinghuai Zhang, Yoshua Bengio. GFlowNets and variational inference. ICLR 2023

**Questions:**

1) Can you please clarify, are the metrics computed over trajectories sampled using $P_F$ or $P_M$ (see Weaknesses)?

2) How are flow estimates $\log F_D$ initialized for the newly added MCDS edges in the hypergrid experiment in Section 4.1 (when there are no neural networks)?

3) Do I understand correctly that MCDS DAG is cleared and rebuilt from scratch every few iterations of training? Could it be beneficial to keep the DAG from previous iterations, instead gradually increasing its size over the course of training?

4) Can the proposed approach be applied when the backward policy is also being trained?

---

> ### Author Response · Authors · 2024-11-22
> **Point Response to Reviewer F11R (Part 1)**
>
> **Comparison with other baselines:**
>
> > Since the authors position the purpose of their approach to be improving trajectory sampling for GFlowNet training, I would like to see some baselines other than the standard on-policy training. E.g. replay buffer approaches [3, 4], and using a mixture of a forward policy and a uniform distribution [5], which were demonstrated to also improve the efficiency of GFlowNet training. As stated by the authors, combining MCDS with such methods could lead to further improvements, but putting some direct comparison to them is also crucial for understanding the efficiency of MCDS.
>
> We have added some experiments about epsilon-exploration in the Hypergrid environment to the Appendix, see Section 7.4. Overall, MCDS still offers an improvement over on-policy.
>
> **L1 metric computation:**
>
> > In addition, I have a concern over which policy is used for evaluation. A standard approach to calculate L1 metric on hypergrids is to use an empirical distribution of terminal states of the last $N$ trajectories sampled for training, and then compare it to the true reward distribution. If this approach is also used here, then the policy being evaluated would be not the GFlowNet forward policy $P_{F}$, but the combination policy $P_{M}$. Then, such evaluation would show the improvements coming not only from introducing a different distribution over trajectories for GFlowNet training (motivation stated by the authors), but also from the adjustment to the forward policy introduced by MCDS. Can the authors please provide more details on how the metric is calculated? For factor graph environment I have the same question.
>
> In both the Hypergrid and Factor Graph experiments, we do not employ search during inference for our MCDS/MCTS evaluations. While it is true that we draw samples from $P_{M}(s'|s)$ to train the models, we re-sample trajectories (independently) using $P_{F}(s'|s;\theta)$ for the purposes of approximating $P(x;\theta)$ to estimate L1 error and JSD. As the reviewer inferred, we maintain a buffer of the previous $N$ sampled trajectories to obtain a running estimate of $P(x;\theta)$. For both Hypergrid and Factor Graph experiments, we use $N=2e5$. In the original submission, we mention this in the Appendix, see Section 7.3.
>
> **Runtime measurements and computational overhead:**
>
> > It will also be good to include some runtime measurements to see how much* of a computational overhead MCDS adds in comparison to standard on-policy training.
>
> The overhead of MCDS depends primarily on the construction budget, the number of parallel workers, and the frequency with which the search DAG is cleared.
>
> Table 4 summarizes the relative slowdown of different MCDS variants used in the Hypergrid and Factor Graph experiments (DB parameterization). Note that the reported metrics include time associated with the calculation of rewards, losses, gradients, and evaluation metrics. With the configurations we tested, the total MCDS runtime penalty ranges from a factor of 1.30 to 3.47.
>
> **Correction to Eq. 3:**
>
> > A minor remark, if $Q(s,s')$ in Equation 3 is an entropy-regularized action value function defined consistently with the standard RL notation, the correct way to write it would be $Q(s,s') = R(s,s') + \log \sum_{s''} \exp Q(s',s'')$. On the other hand, if it is defined as an exponent of action value function, I believe there should be no logarithm in the equation on line 187.
>
> This is correct, we apologize for the error and will update the equation on line 187 to reflect this change.
>
> Equation 3 will read: $Q_{T}(s,s') = R(s,s') + Q_{T}(s)$
>
> The equation on line 187 will read: $Q_{T}(s) = \log \sum_{s' \in Ch(s)} \exp Q_{T}(s,s')$
>
> **Flow estimates for hypergrid in Section 4.1:**
>
> > How are flow estimates $\log F_{D}$ initialized for the newly added MCDS edges in the hypergrid experiment in Section 4.1 (when there are no neural networks)?
>
> We assume the author is referring to the experiments in Figure 2. The flow estimates are all initialized to be zero (meaning the initial policy $P_{D}$ is random uniform). Note that this does not affect the properties of the MCDS or MCTS flows after exhaustive sampling. Some other initialization strategy might be more appropriate (i.e. using a function that depends on the length of the shortest path between the root and the newly added node), but we think this simple strategy is sufficient for a simple demo that is supposed to highlight the differences between MCDS and MCTS at convergence, rather than behaviour of the two algorithms with small sample size.

---

> > ### Author Response · Authors · 2024-11-22
> > **Point Response to Reviewer F11R (Part 2)**
> >
> > **Recomputing vs keeping the MCDS DAG:**
> >
> > > Do I understand correctly that MCDS DAG is cleared and rebuilt from scratch every few iterations of training? Could it be beneficial to keep the DAG from previous iterations, instead gradually increasing its size over the course of training?
> >
> > Yes, and the frequency with which the DAG is cleared is a hyperparameter. It could be helpful to keep growing a single DAG as you suggested. However, in small environments (8x8 Hypergrid) we noticed that as the approximation $P_{M}(s'|s)$ approaches the true distribution $P_{F}(s'|s)$, the sampling becomes less efficient for model training; in fact, it can be outperformed by simple on-policy sampling. This observation is reflected in the experiments of (Atanackovic et al 2024, Figure 5a). Intuitively, for model training it is important to oversample regions where the current terminating distribution $P(x;\theta)$ and the true distribution $P(x)$ differ.
> >
> > **Fixed vs trained backward policy:**
> >
> > > Can the proposed approach be applied when the backward policy is also being trained?
> >
> > Yes: in fact, for the hypergrid experiments we use a learned backward policy, as noted in the Appendix (7.3). However, for the other two experiments we use a fixed uniform backward policy.
> >
> > **References**
> >
> > [Atanackovic et al 2024](https://arxiv.org/abs/2402.05309)

---

> > > ### Comment · Reviewer_F11R · 2024-11-26
> > >
> > > I thank the authors for their answers to my questions and concerns. I decided to raise my score, however, I am still unsure about recommending acceptance at this point, and strongly advise the authors to evaluate on larger environments and directly compare to replay buffer approaches on all environments in future revisions.

---

### Official Review · Reviewer_VkKN · 2024-11-03

**Soundness:** 2
**Presentation:** 2
**Contribution:** 3
**Rating:** 5
**Confidence:** 4

**Summary:**

The authors present a method that combines an MCTS-like algorithm with any GFlowNet training objective to enhance amortized sampler training. This MCTS-type procedure improves the quality of trajectories used during off-policy training, while the training itself can be performed using any existing GFlowNet method. The authors demonstrate the procedure's correctness and validate it through experiments on standard hypergrid and factor graph GFlowNet environments and a language-based reasoning task.

**Strengths:**

- The authors have proposed a new method based on the Monte-Carlo DAG search that can be combined with any GFlowNet training method, contrasting with a concurring MCTS-based method. In particular, this paper extends the work (Buesing et al. 2020) to the case of non-autoregressive generation, which seems non-trivial to me.
- All the theoretical proofs about correctness look valid to me.

Buesing, L., Heess, N., & Weber, T. (2020, June). Approximate inference in discrete distributions with monte carlo tree search and value functions. In International Conference on Artificial Intelligence and Statistics (pp. 624-634). PMLR.

**Weaknesses:**

- The theoretical statements seem to be a trivial corollary of the GFlowNet-RL connection by (Tiapkin et al. 2024) and (Deleu et al. 2024), applied to a GFlowNet problem over the subgraph. Thus, I cannot consider the theoretical contribution as a crucial part of this paper.
- Experimental validation: The hypergrid and factor graph environments seem limited in scale, as the final distribution over terminal states is tractable in both, which was used to compute the success metric. The only large-scale problem presented lacks sufficient details for reproducibility. Specifically, there is no information on how MCDS is applied in the language reasoning task. If MCDS is used on underlying structures rather than at the language level, it could be possible to construct an exhaustive or near-exhaustive DAG, simplifying the language reasoning task to an almost trivial problem. To assess whether this is the case, it is essential to know: (a) the size of the MCDS search and (b) the size of the underlying graphs for the reasoning task. The underlying GFlowNet graph appears relatively small, which could make MCTS-like techniques applied to these structures overly effective. Additionally, there are no comparisons with SFT and RLHF baselines for the reasoning task. Comparing only to an inference-level CoT baseline is not enough, as training data is used. It would be more informative to compare the CoT baseline with few-shot examples included in the training set.

**Questions:**

- Why are the results for TBVar significantly different from those in the original paper (Yu et al., 2024)? The original paper reports an accuracy of 98.41% for 4-step planning, whereas your results show approximately 39.5%. The same issue arises for 6-step planning.
- What is the size of the MCDS search in the factor graph environment?
- What is the reason for using an MCDS and sampling algorithm instead of MCTS and standard RL for the reasoning task? Additionally, what is the final GFlowNet reward function used for this problem? RL methods could address reward maximization directly, so it is unclear why a sampling-based approach is necessary.
- How many training steps or epochs were used for the reasoning task?
- How do you handle race conditions during the parallel construction of the graph by *W* parallel workers? Algorithm 1 outlines the sequential use of workers, yet the text mentions parallel execution (line 233).
- The paper may benefit from referencing existing results on Monte Carlo graph search (e.g., Leurent & Maillard, 2020).


Leurent, E., & Maillard, O. A. (2020, September). Monte-carlo graph search: the value of merging similar states. In Asian Conference on Machine Learning (pp. 577-592). PMLR.

Yu, F., Jiang, L., Kang, H., Hao, S., & Qin, L. (2024). Flow of Reasoning: Efficient Training of LLM Policy with Divergent Thinking. arXiv preprint arXiv:2406.05673.

---

> ### Author Response · Authors · 2024-11-22
> **Point Response to Reviewer VkKN (Part 1)**
>
> **Note on theoretical contribution:**
>
> > The theoretical statements seem to be a trivial corollary of the GFlowNet-RL connection by (Tiapkin et al. 2024) and (Deleu et al. 2024), applied to a GFlowNet problem over the subgraph. Thus, I cannot consider the theoretical contribution as a crucial part of this paper.
>
> We agree that the theoretical results are a corollary of previous connections, although for clarity we emphasize that Theorem 2 applies to all GFlowNet problems with deterministic transitions. The assumption of a DAG-structured environment is standard in most GFlowNet formulations.
>
> **Scale of environments:**
>
> > The hypergrid and factor graph environments seem limited in scale, as the final distribution over terminal states is tractable in both, which was used to compute the success metric.
>
> This is correct, although the sizes of our hypergrid and factor graph environments are consistent with previously published works (Malkin et al 2022, Madan et al 2022, Deleu et al 2024), and are generally accepted as useful GFlowNet benchmarks. As noted below, some of the Factor Graphs experiments can be quite large: for example, the Permuted Chain environment has $|\mathcal{S}| \approx 2\mathrm{e}9$.
>
> > The only large-scale problem presented lacks sufficient details for reproducibility. Specifically, there is no information on how MCDS is applied in the language reasoning task. If MCDS is used on underlying structures rather than at the language level, it could be possible to construct an exhaustive or near-exhaustive DAG, simplifying the language reasoning task to an almost trivial problem. To assess whether this is the case, it is essential to know: (a) the size of the MCDS search and (b) the size of the underlying graphs for the reasoning task. The underlying GFlowNet graph appears relatively small, which could make MCTS-like techniques applied to these structures overly effective.
>
> The reviewer has correctly identified that we perform MCDS search on the underlying reasoning graphs. For the reasoning problems we consider, it is possible to construct an exhaustive search DAG in a reasonable time. However, with the exception of some of the 2-step environments, the budgets we use in training are smaller than what is required for exhaustive search (see Table 3 for details).
>
> Note that we do not apply MCDS for inference (in this experiment or any others), we only use on-policy sampling. Also, in the reasoning task we only evaluate the model on problems (environments) which are not in the training set; exhaustively exploring the training environments would not necessarily lead to perfect performance at inference time.
>
> **Reasoning baselines:**
>
> > Additionally, there are no comparisons with SFT and RLHF baselines for the reasoning task. Comparing only to an inference-level CoT baseline is not enough, as training data is used. It would be more informative to compare the CoT baseline with few-shot examples included in the training set.
>
> If our goal was simply to improve reasoning (by any means), we agree that it would be useful to consider other formulations of the problem that are not using GFlowNets (for example RLHF and DPO, which essentially formulate reasoning as an RL task with a Bradley-Terry reward model). However, we were more interested in demonstrating our method's performance on a challenging GFlowNet benchmark task.

---

> > ### Author Response · Authors · 2024-11-22
> > **Response to Reviewer VkKN (Part 2)**
> >
> > **Results for TBVar:**
> >
> > > Why are the results for TBVar significantly different from those in the original paper (Yu et al., 2024)? The original paper reports an accuracy of 98.41% for 4-step planning, whereas your results show approximately 39.5%. The same issue arises for 6-step planning.
> >
> > There are a number of differences between our setup and the setup used in (Yu et al 2024):
> >
> > - *Backward Policy:* The authors treat the environment as tree-structured (despite its clear DAG structure) and assume that $P_{B}(s|s') = 1$ always. In our opinion this assumption is not justifiable, so we instead use a DAG structure with a uniform $P_{B}(s|s')$.
> > - *Transitions:* The authors use the LLM to model state transitions in addition to the forward policy. The BlocksWorld environment's transition function is deterministic and easy to implement. While it is technically possible to model deterministic transition functions stochastically with GFlowNets (Bengio et al 2023, Appendix C), the authors do not make the necessary algorithmic adjustments to do so. The authors instead use a heuristic approach of sampling transitions from the LLM and checking for validity after the fact; in our opinion this approach is not well-motivated. Regardless, learning transition functions is beyond the scope of our work, so for simplicity we assume the BlocksWorld transition function is known.
> > - *Actions:* The authors modify the action space of the policy function by preventing the same action from being repeated in a trajectory. This violates the Markov assumption of GFlowNets and in our opinion is not justifiable, so we omit this.
> > - *Optimization:* The authors employ multiple schedulers (learning rate, softmax temperature, reward temperature, uniform exploration epsilon, replay buffer sampling probability) and a form of local search \cite{gfn_local_search} to improve training: for simplicity we have removed these.
> > - *ICL examples:* the authors use 4 In-Context Learning (ICL) examples for every prediction; we use 2 to reduce memory and computational overhead (each example is quite large).
> >
> > Because of these differences, we do not believe that it is possible to meaningfully compare our results with those from (Yu et al 2024).
> >
> > **Size of MCDS search in factor-graph experiment:**
> >
> > > What is the size of the MCDS search in the factor graph environment?
> >
> > The MCDS budget is roughly defined as the number of SELECT-EXPAND-BACKUP cycles performed. For the factor graph experiments, we always used a budget of 1024, with 16 parallel workers and a build frequency of 16.
> >
> > Each factor graph environment has $|\mathcal{S}| = 1+(K+1)^N$, $|\mathcal{X}| = K^N$, and $|\mathcal{A}| = K^N + \sum_{n=1}^{N-1} {N \choose n} (N - n) K^{n+1}$.
> >
> > The Permuted Chain environment ($K=5$, $N=12$) has $|\mathcal{S}| = 2176782337 \approx 2\mathrm{e}9$, $|\mathcal{X}| = 244140625 \approx 2\mathrm{e}8$, $|\mathcal{A}| = 22011963985 \approx 2\mathrm{e}10$.
> >
> > The Factorgraphs1 environment ($K=5$, $N=8$) has $|\mathcal{S}| = 1679617 \approx 2\mathrm{e}6$, $|\mathcal{X}| = 390625 \approx 4\mathrm{e}5$, $|\mathcal{A}| = 11588065 \approx 1\mathrm{e}7$.
> >
> > **Reason for using an MCDS and sampling algorithm and clarification on reward function:**
> >
> > > What is the reason for using an MCDS and sampling algorithm instead of MCTS and standard RL for the reasoning task? Additionally, what is the final GFlowNet reward function used for this problem? RL methods could address reward maximization directly, so it is unclear why a sampling-based approach is necessary.
> >
> > We are mainly interested in using the reasoning task as a challenging GFlowNet benchmark. We do not claim that formulating the problem in this way (i.e. instead of using standard RL) is necessary for achieving good performance. Other recent works have highlighted the advantages of using GFlowNets for language modeling and reasoning tasks (Hu et al 2023), arguing that these methods can offer improved diversity over their RL counterparts.
> >
> > **Number of steps for reasoning task:**
> >
> > > How many training steps or epochs were used for the reasoning task?
> >
> > There isn't really a notion of epochs for training these types of models, since training simply involves drawing samples from an environment (there is no fixed dataset).
> >
> > 20 gradient updates in total (each using a batch size of 4) were performed during training. The reasoning problem that defines the environment was sampled independently (with replacement) for each training iteration. This is similar to the setup used in (Yu et al 2024).

---

> > > ### Author Response · Authors · 2024-11-22
> > > **Response to Reviewer VkKN (Part 3)**
> > >
> > > **Race conditions and parallel construction of workers:**
> > >
> > > > How do you handle race conditions during the parallel construction of the graph by W parallel workers? Algorithm 1 outlines the sequential use of workers, yet the text mentions parallel execution (line 233).
> > >
> > > All of our environments are vectorizable, in the sense that states/actions can be represented as vectors and transitions can be represented with linear algebra operations. This allows efficient batching of all tree construction operations (i.e. using tensor stacking and broadcasting).
> > >
> > > Logically, constructing the tree with a batch size of $W$ is equivalent to using $W$ workers operating in lockstep. In each step of the algorithm (Select, Expand, Backup) all workers wait (block) until they are all complete before advancing to the next step.
> > >
> > > The Select step does not have any race conditions, since each worker's trajectory evolves independently. In the Expand step, if two workers or more workers try to expand the same state, it is only expanded once. In the Backup step, if two or more workers update the same state, the updates are averaged.
> > >
> > > **References**
> > >
> > > [Bengio et al 2021](https://arxiv.org/pdf/2111.09266)
> > >
> > > [Malkin et al 2022](https://arxiv.org/pdf/2201.13259)
> > >
> > > [Madan et al 2022](https://arxiv.org/pdf/2209.12782)
> > >
> > > [Deleu et al 2024](https://arxiv.org/pdf/2402.10309)
> > >
> > > [Yu et al 2024](https://arxiv.org/pdf/2406.05673)
> > >
> > > [Hu et al 2024](https://arxiv.org/pdf/2310.04363)

---

> > > > ### Comment · Reviewer_VkKN · 2024-11-27
> > > >
> > > > I would like to thank the authors for their response. I am still very concerned about the experimental setup for the reasoning problem.
> > > >
> > > > 1. Performing operations on the underlying graph structures seems to be a form of cheating because the task-agnostic methods (such as TBVar or any SFT/RLHF method) operate only on the level of language. In contrast, the presented method requires strong knowledge about the underlying problem. In particular, this approach might be difficult to generalize to other reasoning problems (especially to the less artificial, such as reasoning in math).
> > > > 2. As you mentioned, you removed the local search component from the TBVar method, which plays a very important role. Additionally, as other reviewers mentioned, the MCDS is not the only way to generate more high-quality data for training in the GFlowNet literature: (prioritized) experience replay buffers, local search, and eps-greedy exploration. All these components were important for the baseline algorithm, and in your study, you have removed all these components. Overall, I cannot consider the comparison with TBVar fair.
> > > > 3. Regarding the generation budget, I found that using 100 search iterations in the case of the maximal number of states of about 400 should make the generated data very close to ground truth, and it provides enormous benefits for the method. I would consider the authors to provide an ablation study on the MCDS budget for the reasoning task.
> > > > 4. Additionally, I started doubting that this environment might be considered a large-scale one since it used only 20 gradient steps to solve. Also, I am not sure that 20 gradient steps are enough to estimate the flow function for SubTB or DB methods since the flow head is typically initialized randomly (whereas TBVar avoids this problem as the method requires training only the policy itself, which benefits from the pretraining).
> > > >
> > > > Overall, I like the idea of the paper, but it requires more empirical work to show its benefits, e.g., comparing it to local search / prioritized replay buffer methods, studying the influence of the search depth, etc. I would suggest the authors provide an evaluation of their method on other reasoning benchmarks, presented in the work of (Yu et al. 2024), or consider more classical GFlowNet benchmarks, such as molecule generation (Bengio et al. 2021) or phylogenetic tree generation (Zhou et al. 2023).
> > > >
> > > > ### References
> > > >
> > > > Bengio, E., Jain, M., Korablyov, M., Precup, D., & Bengio, Y. (2021). Flow network based generative models for non-iterative diverse candidate generation. Advances in Neural Information Processing Systems, 34, 27381-27394.
> > > >
> > > > Zhou, Mingyang, et al. "PhyloGFN: Phylogenetic inference with generative flow networks." arXiv preprint arXiv:2310.08774 (2023).

---

> > > > > ### Author Response · Authors · 2024-11-27
> > > > >
> > > > > We thank the reviewer for their comments, and include come clarifications below:
> > > > >
> > > > > > Performing operations on the underlying graph structures seems to be a form of cheating because the task-agnostic methods (such as TBVar or any SFT/RLHF method) operate only on the level of language. In contrast, the presented method requires strong knowledge about the underlying problem. In particular, this approach might be difficult to generalize to other reasoning problems (especially to the less artificial, such as reasoning in math).
> > > > >
> > > > > To be clear, the experiments from Yu et al 2024 also use the underlying graph structures (i.e. for determining the set of possible actions in each state, in the local search, etc). We agree that comparisons with SFT/RLHF that don't have access to the underlying structures would be unfair, which is one of the reasons we didn't include them.
> > > > >
> > > > > All of our baseline methods (including TBVar) have access to the underlying structure. Again, we are not making a claim that MCDS is a SOTA method for reasoning in an LLM, rather we view this formulation of the BlocksWorld task as a challenging GFlowNet benchmark that involves a large model.
> > > > >
> > > > > > As you mentioned, you removed the local search component from the TBVar method, which plays a very important role. Additionally, as other reviewers mentioned, the MCDS is not the only way to generate more high-quality data for training in the GFlowNet literature: (prioritized) experience replay buffers, local search, and eps-greedy exploration. All these components were important for the baseline algorithm, and in your study, you have removed all these components. Overall, I cannot consider the comparison with TBVar fair.
> > > > >
> > > > > As we mentioned, our setup is different from Yu et al 2024 in many ways (including the actual definition of the environment), for the reasons we outlined in the previous response. Our goal is not to explicitly compare with what they were doing, and we are not claiming that the TBVar in our experiment is comparable to the method presented in Yu et al 2024. We simply used the setup in the paper to inspire our own environment.
> > > > >
> > > > > > Regarding the generation budget, I found that using 100 search iterations in the case of the maximal number of states of about 400 should make the generated data very close to ground truth, and it provides enormous benefits for the method. I would consider the authors to provide an ablation study on the MCDS budget for the reasoning task.
> > > > >
> > > > > This is a fair point, however we would note two things:
> > > > >
> > > > > 1) Regrettably, in Table 3 we accidentaly swapped two columns for the 8-step experiment: the entries for $|\mathcal{X}|$ and $|\mathcal{S}|$. This is evident because $\mathcal{X} \subseteq \mathcal{S}$ by definition. In fact, the maximum number of states in the 8-step environment is 1232. We apologize for the confusion.
> > > > >
> > > > > 2) We would argue that it's more relevant to compare the size of the edge set $|\mathcal{A}|$ with the budget.
> > > > >
> > > > > > Additionally, I started doubting that this environment might be considered a large-scale one since it used only 20 gradient steps to solve. Also, I am not sure that 20 gradient steps are enough to estimate the flow function for SubTB or DB methods since the flow head is typically initialized randomly (whereas TBVar avoids this problem as the method requires training only the policy itself, which benefits from the pretraining).
> > > > >
> > > > > The flow head shares parameters with the policy (i.e. the LLM), only a small linear head is randomly initialized. In this way it has a strong (language-based) prior.
> > > > >
> > > > > **References**
> > > > >
> > > > > [Yu et al 2024](https://arxiv.org/pdf/2406.05673)

---

### Official Review · Reviewer_tpNn · 2024-11-04

**Soundness:** 2
**Presentation:** 2
**Contribution:** 2
**Rating:** 6
**Confidence:** 3

**Summary:**

Generative flow networks (GFlowNets) are models for sequentially generating combinatorial objects. GFlowNets are trained by sampling trajectories from the network and optimizing some loss function with respect to the trajectory outputs. This paper proposes a Monte Carlo tree search (MCTS) -based approach to sampling trajectories for training GFlowNets. They apply this approach to different types of GFlowNets trained on different objectives and compare the performance against the conventional on-policy sampling strategy.

**Strengths:**

**Originality:** Whiles MCTS and GFlowNets are not novel ideas, the novelty arises in the adaptation of MCTS sampling to a domain in which it has not been used, namely, in sampling trajectories for training GFlowNets.

**Significance:** MCTS is a popular, easily-implementable algorithm known to improve sampling efficiency in planning domains and provide statistical guarantees on performance. Having an MCTS-based sampling method for training GFlowNets can be a quick-and-easy approach for improving on the performance of these models that can be adopted by anyone working with GFlowNets.

**Clarity:** Not being familiar with prior work in GFlowNets, it did not take too much effort to get the gist of how they work from this paper. That being said there are some concerns with the clarity (described in the Weaknesses.)

**Quality:** For the most part the approach for adapting MCTS to GFlowNets (which the authors call MCDS) is intuitively sound. The authors also chose a sufficient number of domains (one demonstrative and two complex) for conceptualizing GFlowNets (with and without MCDS) and measuring its performance.

**Weaknesses:**

While the paper presents an interesting and simple idea, there are a number of concerns with the work. The more concerning issues deal with the experimental evaluation.

1. The results for Hypergrid (Fig 3) do not appear meaningful:

    - The plot states that it shows mean and standard deviation however there no markers, fills or any visual indication of the standard deviation.

    - The standard deviation is not a statistical significance test --- it is a measure of dispersion. Confidence intervals would be an appropriate choice for visualizing statistical significance.

    - Without any statistical testing it is difficult to verify whether the difference in performances are significant. Hence, I believe the results do the support the authors' claims that MCDS improves the performance of GFlowNets over on-policy sampling in Hypergrid.

    - Secondly, inspecting the plots do not show that their is indeed any performance gain over conventional MCTS sampling.

    - Lastly, even if CI plots were produced it is hard to imagine that averaging over a mere 3 seeds would produce results of significance. With 3 seeds the authors may just plot the results over each seed.

2. Some of the issues from (1) similarly exist for the results in Table 1. The standard deviations are reported however this provides no indication of statistical significance.

3. The explanation of the Hypergrid domain in Sec 5.1 did not make any sense and there was no explanation of the performance metric. I had to read the original GFlowNet paper to understand it.

4. The explanation for MCTS in Sec 2.2 describes it such that all states in a MCTS-tree, and consequently MCDS-tree, are unique which is not necessarily true. In fact, if I understand the diagrams in Fig 2, the equivalent trees all will have duplicate states.

5. Additionally, in Fig 2, I cannot understand why the forward policy for MCTS is the way it is. If the reward function for MCTS is using the one shown in Eq 2, then if sampled enough, MCTS should terminate at the bottom left corner and observe a reward. Am I missing something?

**Questions:**

Line 57: "benchmark" has a typo

Line 157: What is $Q$? It hasn't been defined.

---

> ### Author Response · Authors · 2024-11-22
> **Point Response to Reviewer tpNn**
>
> **Standard Deviation in the plots:**
>
> > The plot states that it shows mean and standard deviation however there no markers, fills or any visual indication of the standard deviation.
>
> The standard deviations are plotted in Figure 3, but they are small so it is difficult to see.
>
> > The standard deviation is not a statistical significance test --- it is a measure of dispersion. Confidence intervals would be an appropriate choice for visualizing statistical significance. Without any statistical testing it is difficult to verify whether the difference in performances are significant. Hence, I believe the results do the support the authors' claims that MCDS improves the performance of GFlowNets over on-policy sampling in Hypergrid.
>
> Statistical testing is only one tool to establish significance in an experiment. We chose to plot the standard deviation because it can give an idea of the variability of the compared algorithms, not just the difference in average performance. The reviewer notes that the standard deviations are too narrow to see; it would be even narrower if we plotted standard error. If we increased the sample size, the standard error would also get narrower.
>
> **Comparision to MCTS sampling:**
>
> > Secondly, inspecting the plots do not show that their is indeed any performance gain over conventional MCTS sampling.
>
> Please see the curves in Figure 3c and 3f in which the MCTS curves overlap with the on-policy curves, while the MCDS curves in Figure 3a and 3d descend faster than the on-policy curves, hence showing improvement.
>
> **Number of seeds:**
>
> > Lastly, even if CI plots were produced it is hard to imagine that averaging over a mere 3 seeds would produce results of significance. With 3 seeds the authors may just plot the results over each seed.
>
> We did not see much variation for the runs with different seeds for our method, and hence we do not expect the plots for Hypergrid to change by adding more than 3 seeds. Please note that for our factor graph experiments (Figure 4), where the results are much more variable, we included more seeds (at least 10) and plotted individual seeds as dashed lines.
>
> **Details of the Hypergrid domain:**
>
> > The explanation of the Hypergrid domain in Sec 5.1 did not make any sense and there was no explanation of the performance metric. I had to read the original GFlowNet paper to understand it.
>
> We apologize for the confusion. Hypergrid is well-known in the GFlowNet community, although it might be confusing for readers who are less familiar with this area. We will elaborate on the definition in the Appendix (7.3).
>
> **Clarification on Fig 2 and MCTS-tree and terminating distribution:**
>
> > The explanation for MCTS in Sec 2.2 describes it such that all states in a MCTS-tree, and consequently MCDS-tree, are unique which is not necessarily true. In fact, if I understand the diagrams in Fig 2, the equivalent trees all will have duplicate states.
>
> The version of MCTS we are describing is similar to the one described in Buesing et al 2019. In this setting (and the GFlowNet setting), we assume that all states in the tree are unique (even though it might be possible to reach them with multiple paths). This assumption of uniqueness also applies to MCDS. In Figure 2, each state in the environment is a cell in the 2-dimensional hypergrid, so there are no duplicate states.
>
> **Terminating distribution:**
>
> > Additionally, in Fig 2, I cannot understand why the forward policy for MCTS is the way it is. If the reward function for MCTS is using the one shown in Eq 2, then if sampled enough, MCTS should terminate at the bottom left corner and observe a reward. Am I missing something?
>
> Because we are using maximum entropy MCTS, the terminating distribution $P_{D}(x)$ is biased by the number of trajectories leading to each state $x$ (see Bengio et al 2021 for more details).
>
> The number of paths from the initial state $s_0 = (0,0)$ (bottom-left) and an arbitrary state $s_{m,n} = (m,n)$, where $m,n \leq 7$, is equal to the binomial coefficient ${m+n \choose n} = \frac{(m+n)!}{m!n!}$. Consider two modes in the 8x8 Hypergrid: the mode at $(1,1)$ and the mode at $(6,6)$. The former only has $\frac{2!}{1!} = 2$ trajectories that lead to it, while the mode at $(6,6)$ has $\frac{12!}{6!6!} = 924$ trajectories that lead to it. The MCTS solution (i.e. the maximum entropy solution) greatly favours termination at states that are further from the initial state $s_0$, because these states can be reached with a greater number of trajectories.
>
> **Definition of Q:**
>
> > Line 157: What is $Q$? It hasn't been defined.
>
> It is defined on Line 151 in Section 2.2 (of the original paper).
>
> **References**
>
> [Bengio et al 2021](https://arxiv.org/pdf/2111.09266)
>
> [Buesing et al 2019](https://arxiv.org/pdf/1910.06862)

---

> > ### Comment · Reviewer_tpNn · 2024-11-26
> > **Acknowledgement of rebuttal**
> >
> > I thank the authors for their rebuttal. I think the authors address my main concerns. I will raise my score.

---

### Official Review · Reviewer_YCcQ · 2024-11-14

**Soundness:** 2
**Presentation:** 1
**Contribution:** 3
**Rating:** 3
**Confidence:** 3

**Summary:**

This paper introduces the Monte Carlo DAG search (MCDS) algorithm, a modified MCTS method for sampling trajectories for GFlowNet training. The paper proves MCDS can find optimal flows for a GFlowNet given exhaustive samples from the environment. The algorithm is evaluated on three discrete domains, and some of the results show improved performance over on-policy sampling.


My recommendation for the article is reject for three main reasons:
1. Poor presentation of background and method.
2. Overclaiming about results of experiments.
3. Poor empirical practices and missing baselines.

**Strengths:**

1. The idea for MCDS is an interesting combination of ideas from MaxEnt RL and MCTS and the overall method is well motivated.
2. The problem of improving GFlowNets is relevant and well positioned by prior work.
2. Theorem 2 and its proof are novel and presented clearly.
3. The introduction is clear and well-written.

**Weaknesses:**

1. The background and method sections are poorly presented and are hard to follow. There is too much notation. The text is not coherent.

2. Some claims are not supported by the experiment results, which is exacerbated by poor empirical practices. For instance, line 411: "MCDS improves training compared to on-policy sampling" is supported by the learning curves in Figure 3. However, these curves are averaged over 3 runs, and the shaded regions (which are not visible) are standard deviations, a measure of dispersion and not confidence regarding the mean. There is little evidence that the difference is statistically significant. Standard deviation is not a measure of confidence and 3 seeds are insufficient to make such a general claim. The choice of hyperparameters is not justified. The reported metric is also not justified. The same criticism also applies to line 412: "Larger tree construction budgets providing a bigger improvement".

3. In general, the hyperparameter choices for no experiment are justified, and all claims regarding differences between methods in Hypergrid and Blocksworld are not supported by statistical tests (e.g. confidence measures).

4. While the experiments show comparisons with on-policy sampling, there is no comparison with other baselines that attempt to improve GFlowNet training, such as epsilon uniform exploration, replay buffers, or local search. This limitation is acknowledged in the paper, however, I think their inclusion is required to evaluate MCDS.

5. Minor edits:

  - Line 057: typo "benchmarkpr".
  - Line 325: repeated "performance performance".
  - The reported metrics should be motivated and introduced.

**Questions:**

1. What does the top plot in Figure 2 show? What do the lines represent? Why is this difference significant?
2. How were the hyperparameters selected? Why?

---

> ### Author Response · Authors · 2024-11-22
> **Point Response to Reviewer YCcQ**
>
> **Experimental results, seeds and std deviation:**
>
> > Some claims are not supported by the experiment results, which is exacerbated by poor empirical practices. For instance, line 411: "MCDS improves training compared to on-policy sampling" is supported by the learning curves in Figure 3. However, these curves are averaged over 3 runs, and the shaded regions (which are not visible) are standard deviations, a measure of dispersion and not confidence regarding the mean. There is little evidence that the difference is statistically significant. Standard deviation is not a measure of confidence and 3 seeds are insufficient to make such a general claim. The choice of hyperparameters is not justified. The reported metric is also not justified. The same criticism also applies to line 412: "Larger tree construction budgets providing a bigger improvement".
>
> Our presentation of the Hypergrid results in Figure 3 are consistent with previous works (Bengio et al 2021, Madan et al 2022). Since Hypergrid is an easier environment, the difference in the learned L1-errors are quite low for our method. We don't expect the plots to change by adding more seeds or by using other metrics.
>
> **Hyperparameter selection:**
>
> > In general, the hyperparameter choices for no experiment are justified, and all claims regarding differences between methods in Hypergrid and Blocksworld are not supported by statistical tests (e.g. confidence measures).
>
> We have added more information about the hyperparameters to the Appendix, thank you for this suggestion.
>
> **Comparison with other baselines:**
>
> > While the experiments show comparisons with on-policy sampling, there is no comparison with other baselines that attempt to improve GFlowNet training, such as epsilon uniform exploration, replay buffers, or local search. This limitation is acknowledged in the paper, however, I think their inclusion is required to evaluate MCDS.
>
> We've added some experiments regarding epsilon uniform exploration in the Hypergrid to the appendix, see Section 7.4. Overall, MCDS still offers an improvement over on-policy.
>
> **Clarification on Figure 2:**
>
> > What does the top plot in Figure 2 show? What do the lines represent? Why is this difference significant?
>
> The top plot represents the approximation quality of $P_{D}(x)$ induced by flows $F_{D}(s,s')$ calculated with MCDS and maximum entropy MCTS. The line plot shows approximation error (measured using Jensen-Shannon divergence) with different tree construction budgets. The heat maps visualize the state flows $F_{D}(s)$ in green (darker = larger flow), where each state $s$ is represented as a state in the grid. The policies $P_{D}(s'|s)$ are visualized with blue arrows and red dots. In the visualization, the probability of each action is proportional to the size of its corresponding symbol.
>
> In an 8x8 hypergrid, which has $|\mathcal{S}| = 65$ and $|\mathcal{A}| = 176$, a budget of 176 should theoretically be sufficient for exhaustive exploration with MCTS/MCDS. However, with our parallel implementation there is a stochastic component: when the number of workers is greater than one, it is possible for multiple workers to "EXPAND" the same node, potentially resulting in an incomplete tree even after the budget of 176 is exhausted. For this experiment we confirmed that budgets of $2^9$ and $2^{10}$ resulted in exhaustive exploration.
>
> Both methods produce better approximations with larger budgets, but MCTS is unable to converge to the correct solution (i.e. JSD of zero) even with a very large budget. This happens because maximum entropy MCTS results in a flows that are biased towards terminating states with a larger number of trajectories. This is highlighted by the fact that in the four squares in the bottom-left corner, the "terminate" action receives very little probability mass in the MCTS visualization when compared to MCDS/optimal. These four states actually have high reward, but the MCTS solution assigns low probability of terminating because the trajectory lengths to get to them are very short (less than 2 steps).
>
> To see this more clearly, we've modified Figure 2 in the updated paper to show the predicted terminating distributions $P_{D}(x)$ for the tree policies from MCTS and MCDS, instead of the state flows $F_{D}(s)$.
>
> Note that in the hypergrid domain, every state is a terminating state. Also, this domain is small enough such that the optimal flows and policies, and the terminating distribution, can be calculated exhaustively in a short time. We also assume that there is a uniform backward policy $P_{B}(s|s')$ as is commonly used for GFlowNets, although this demonstration would also work with a different choices of $P_{B}(s|s')$.
>
> **References**
>
> [Bengio et al 2021](https://arxiv.org/abs/2106.04399)
>
> [Madan et al 2022](https://arxiv.org/pdf/2209.12782)

---

> > ### Author Response · Authors · 2024-11-28
> > **Gentle Reminder**
> >
> > Dear Reviewer YCcQ,
> >
> > We have carefully replied to your initial concerns in our rebuttal. We invite you to review our response and updated manuscript.
> >
> > If our response is satisfactory, we kindly request that you reconsider your score. We remain open to further discussion to ensure a fair and thorough review process.
> >
> > Thank you for your time and expertise.

---

> > ### Comment · Reviewer_YCcQ · 2024-12-02
> >
> > I thank the authors for their rebuttal. The updated manuscript does a better job of explaining the work. However, no modifications or reconsiderations of claims are made in the paper to warrant increasing the score. I maintain my original assessment.

---

### Author Response · Authors · 2024-11-22
**Author Rebuttal**

We thank all reviewers for their valuable feedback. We have uploaded a modified version of the paper with changes highlighted in blue. We provide point-by-point responses to each question and refer to parts of the updated paper where applicable.

---

### Author Response · Authors · 2024-11-26
**Summary**

We thank the reviewers for their constructive feedback and suggestions.

To summarize our rebuttal:

- We've added experiments regarding epsilon-uniform exploration that strengthen our results
- We've clarified information about experiments (including environment definitions and hyperparameter settings)
- We've modified Figure 2 to visualized the terminating distribution P(x) instead of the state flows F(s), to better highlight the difference between MCTS and MCDS
- We've included some timing experiments

Please let us know if there are any other pending questions or concerns.

---

### Meta-Review · Area_Chair_a2Xv · 2024-12-20

**Metareview:**

This paper proposes to apply monte-carlo tree search with reinforcement learning to improve GFlowNet training. However, as noted by the reviewers, there are serious method logical flaws in the experiments that call into question the paper's results. I suggest the authors revise the design of the experiments to address the reviewers' concerns.

**Additional Comments On Reviewer Discussion:**

There was back and forth where the authors and reviewers discussed the experiments and theoretical results. However, multiple reviewers did not have their concerns addressed in this discussion. The lack of empirical rigor is the main reason for rejecting this paper.

---

### Decision · Program_Chairs · 2025-01-22

Reject